# KL-VS heterozygosity is associated with lower amyloid-dependent tau accumulation and memory impairment in Alzheimer's disease

Julia Neitzel [1,2,3 ✉], Nicolai Franzmeier [1], Anna Rubinski[1], Martin Dichgans [1,4,5], Matthias Brendel [6], Alzheimer's Disease Neuroimaging Initiative (ADNI)*, Rainer Malik[1] & Michael Ewers [1,4 ✉]

Klotho-VS heterozygosity (KL-VS[het]) is associated with reduced risk of Alzheimer's disease (AD). However, whether KL-VS[het] is associated with lower levels of pathologic tau, i.e., the key AD pathology driving neurodegeneration and cognitive decline, is unknown. Here, we assessed the interaction between KL-VS[het] and levels of beta-amyloid, a key driver of tau pathology, on the levels of PET-assessed neurofibrillary tau in 551 controls and patients across the AD continuum. KL-VS[het] showed lower cross-sectional and longitudinal increase in tau-PET per unit increase in amyloid-PET when compared to that of non-carriers. This association of KL-VS[het] on tau-PET was stronger in Klotho mRNA-expressing brain regions mapped onto a gene expression atlas. KL-VS[het] was related to better memory functions in amyloid-positive participants and this association was mediated by lower tau-PET. Amyloid-PET levels did not differ between KL-VS[het] carriers versus non-carriers. Together, our findings provide evidence to suggest a protective role of KL-VS[het] against amyloid-related tau pathology and tau-related memory impairments in elderly humans at risk of AD dementia.

[1] Institute for Stroke and Dementia Research, Klinikum der Universität München, Ludwig-Maximilians-Universität LMU, Munich, Germany. [2] Department of Radiology and Nuclear Medicine, Erasmus University Medical Center, Rotterdam, the Netherlands. [3] Department of Epidemiology, Erasmus University Medical Center, Rotterdam, the Netherlands. [4] German Center for Neurodegenerative Diseases (DZNE), Munich, Germany. [5] Munich Cluster for Systems Neurology (SyNergy), Munich, Germany. [6] Department of Nuclear Medicine, Klinikum der Universität München, Ludwig-Maximilians-Universität LMU, Munich, Germany. *A list of authors and their affiliations appears at the end of the paper. ✉email: j.neitzel@erasmusmc.nl; Michael.Ewers@med.uni-muenchen.de

Klotho is a transmembrane protein that has been associated with enhanced longevity and better brain health in aging[1,2]. Klotho is expressed primarily in the kidney and brain, where it has been implicated in a number of vital cellular functions (for review see[3]). Loss-of-function mutations in transgenic mice are associated with reduced Klotho protein expression, accelerated aging phenotypes, and dramatically shortened life span[1,4]. In humans, two variants in the *Klotho* gene (KL, 13q13.1), rs9536314 (F352V) and rs9527025 (C370S), form a functional haplotype. Carrying one copy, but not two copies of the KL-VS haplotype, referred to as KL-VS heterozygosity (KL-VS[het]), has been previously linked to increased Klotho levels in the blood[5,6]. KL-VS[het] occurs in about 20–25% of the population[5] and is associated with higher cognitive performance across the adult life span[5,7–9], larger frontotemporal gray matter volume in cognitively normal individuals[8], and lower mortality[6,10]. Together, these results suggest a crucial role of Klotho in the maintenance of cognitive abilities and brain integrity during aging.

Beyond the protective role of Klotho in normal aging, recent studies suggest an association between Klotho and reduced risk of Alzheimer's disease (AD)[11], the most frequent cause of dementia in the elderly[12]. A recent meta-analysis reported that KL-VS[het] was associated with reduced AD dementia risk and cognitive decline in elderly individuals carrying the ApoE ε4 allele[13], i.e., the strongest genetic risk factor for AD dementia possibly through elevated levels of primary AD pathology including cortical beta-amyloid (Aβ) aggregation[14,15]. Importantly, KL-VS[het] was associated with reduced biomarker levels of Aβ deposition in ApoE ε4 carriers[16] suggesting that KL-VS[het] may directly alter levels of primary AD pathology.

Yet, an open question is whether Klotho is associated with altered levels of fibrillary tangles containing pathologic tau, i.e., the key driver of disease progression in AD[17]. In the presence of Aβ deposition, i.e., the earliest primary pathology in AD[18,19], neurofibrillary tangles spread from the medial temporal lobe to higher cortical areas[20–22]. The progressive development of neurofibrillary tangles in the presence of Aβ pathology is closely associated with gray matter atrophy[23–25] and cognitive worsening[20,26–28] and is more predictive of such alterations than

Aβ[29]. Due to the high clinical relevance of tau pathology, it is pivotal to understand whether the KL-VS[het] variant attenuates the accumulation of neurofibrillary tangles at a given level of Aβ deposition, and thus a cognitive decline in AD. Studies in mouse models of Aβ and accelerated aging reported that enhancing *KL* expression was associated with reduced Aβ burden and phosphorylated tau[11,30], although conflicting results were reported as well[31]. However, these mouse models fail to develop neurofibrillary tangles in the presence of Aβ and thus only incompletely recapitulate AD-specific tau pathology in humans.

Here, we examined whether KL-VS[het] attenuates the association between higher Aβ and higher fibrillar tau assessed via positron emission tomography (PET) in a group of 551 elderly asymptomatic and symptomatic individuals recruited within a large North American multicenter study on AD[32]. We found the KL-VS[het] variant to be associated with an attenuated increase in regional tau-PET at pathological levels of global amyloid-PET, suggesting that KL-VS[het] was potentially protective against Aβ-related increase in neurofibrillary tangles. This association was pronounced in ApoE ε4 carriers. The strength of the KL-VS[het] effect on region-specific tau-PET levels was correlated with the regional expression pattern of *KL* in the brain[33,34] supporting the notion that the KL-VS[het] variant modulates the regional accumulation of tau pathology. Importantly, KL-VS[het] was associated with higher memory performance and this association was mediated by reduced tau-PET levels in KL-VS[het] carriers with the elevated amyloid-PET burden. For Aβ, we did not find the previously reported association between KL-VS[het] and lower Aβ pathology in the current sample, but confirmed this link in a larger sample including all individuals with amyloid-PET but not necessarily tau-PET assessment available indicating that the effect size of KL-VS[het] on Aβ was smaller than that on tau pathology.

## Results

Detailed sample characteristics are presented in Table 1. Among the 551 participants (347 CN, 156 MCI, 48 ADD), there were 144 KL-VS[het] carriers and 407 non-carriers. Demographics (age, sex, and education) or ApoE ε4 status did not differ between KL-VS[het] carrier and non-carrier groups (all $p > 0.05$). Continuous values of global

### Table 1 Sample characteristics.

|  | KL-VS[het] carriers | KL-VS[het] non-carriers | *p* value |
|---|---|---|---|
| *ADNI, all* |  |  |  |
| N | 144 | 407 |  |
| Age | 71.29 (6.61) | 71.39 (6.72) | 0.880 |
| Sex, F:M | 76:68 | 206:201 | 0.727 |
| Diagnosis, CN:MCI:ADD | 102:34:8 | 245:122:40 | 0.059 |
| Education, years | 16.20 (2.50) | 16.65 (2.51) | 0.065 |
| MMSE | 28.17 (2.43) | 28.11 (2.82) | 0.819 |
| ApoE ε4 status, neg:pos | 90:54 | 255:152 | 1.000 |
| Global amyloid-PET, CL | 28.85 (37.94) | 32.28 (40.30) | 0.373 |
| Amyloid-PET status, neg:pos | 89:55 | 232:175 | 0.365 |
| *Longitudinal subsample* |  |  |  |
| N | 52 | 148 |  |
| Age | 70.93 (5.76) | 71.43 (6.56) | 0.631 |
| Sex, F:M | 28:24 | 69:79 | 0.462 |
| Diagnosis, CN:MCI:ADD | 36:12:4 | 77:55:16 | 0.097 |
| Education, years | 15.85 (2.58) | 16.56 (2.51) | 0.081 |
| MMSE | 28.00 (2.59) | 27.92 (2.48) | 0.841 |
| ApoE ε4 status, neg:pos | 25:27 | 81:67 | 0.506 |
| Global amyloid-PET, CL | 45.01 (41.44) | 42.73 (42.59) | 0.739 |
| Amyloid-PET status, neg:pos | 22:30 | 61:87 | 1.000 |
| Tau-PET follow-up, years | 1.54 (0.75) | 1.66 (0.80) | 0.330 |

*CN* cognitively normal, *MCI* mild cognitive impairment, *ADD* Alzheimer's disease dementia, *F* female, *M* male, *MMSE* Mini-Mental State Examination, *neg* negative, *pos* positive, *CL* centiloid.

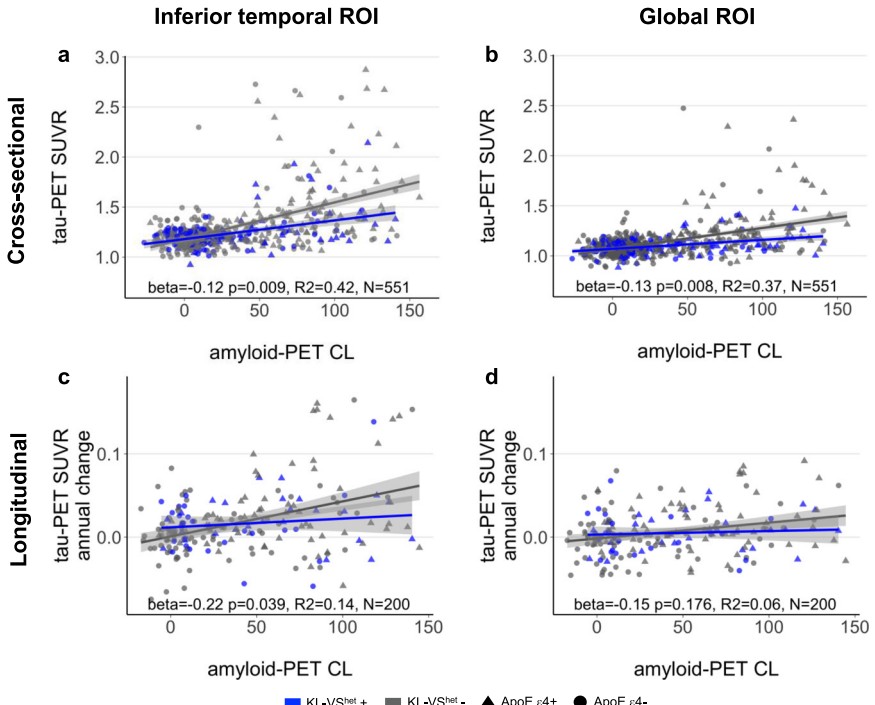

**Fig. 1 Association between KL-VS heterozygosity, amyloid-, and tau-PET.** Scatterplots display the relationship between global amyloid-PET levels and **a**, **b** cross-sectionally assessed tau-PET levels or **c**, **d** longitudinally assessed tau-PET annual change rates measured in inferior temporal gyri (left panel) and globally in neocortical areas (right panel) as a function of KL-VS[het] variant. Blue and gray colors indicate individuals with heterozygous or non-heterozygous KL-VS alleles. Statistics of the KL-VS[het] × amyloid-PET interaction effect on tau-PET uptake were derived from multiple linear regression analyses, controlled for the main effects of KL-VS[het] and amyloid-PET levels as well as age, sex, diagnosis, education, and ApoE ε4 carrier status. Linear model fits are indicated together with 95% confidence intervals.

amyloid-PET uptake did not differ between KL-VS[het] carriers versus non-carriers ($t(265.06) = 0.92$, $p = 0.373$).

**KL-VS heterozygosity is associated with lower amyloid-related tau accumulation.** In the main analysis, we tested the hypothesis that KL-VS[het] modifies the association between Aβ and tau pathology (both assessed by continuous measures of PET uptake). In a region-of-interest (ROI)-based analysis, we focused on tau-PET in the inferior temporal cortex (i.e., ROI of early Aβ-related tau pathology[20,26,35–37]) and whole-brain tau-PET levels. The results of a linear regression analysis showed a significant KL-VS[het] × amyloid-PET interaction effect on tau-PET levels in both the inferior temporal ROI (standardized beta = −0.12, $p = 0.009$, $N = 551$, effect size measured by Cohen's $f = 0.112$) and the global ROI (beta = −0.13, $p = 0.008$, $N = 551$, Cohen's $f = 0.114$). For both tau-PET ROIs, the increase in tau-PET as a function of rising global amyloid-PET was attenuated in KL-VS[het] carriers versus non-carriers (Fig. 1a, b). The main effects of amyloid- on tau-PET for KL-VS[het] carriers and non-carriers are reported in Supplementary Table 1. All analyses were controlled for the main effects of the interaction terms, age, sex, diagnosis, education, and ApoE ε4 carrier status.

Next, we performed several secondary analyses. To ensure that our results were not driven by unequally sized KL-VS[het] groups, we repeated the main analyses in all 144 KL-VS[het] carriers and 144 out of 407 non-carriers who were selected based on propensity score matching for global amyloid-PET levels and diagnosis. Comparable KL-VS[het] × amyloid-PET interaction effects were found on tau-PET levels in both ROIs (inferior temporal ROI: beta = −0.20, $p = 0.005$, $N = 288$, Cohen's $f = 0.168$; global ROI: beta = −0.21, $p = 0.010$, $N = 288$, Cohen's $f = 0.156$; Supplementary Fig. 1a, b). In addition, we repeated the interaction analysis in 1000 bootstrapped samples (i.e., random

sampling from the participant pool with replacement). As a reference, we also generated a null distribution by randomly reshuffling the KL-VS[het] labels on each iteration. The bootstrapped mean $t$-value of the interaction effect differed significantly from that of the null distribution (inferior temporal ROI: $t(1901.9) = −47.43$, $p < 0.001$; global ROI: $t(1839.2) = −48.99$, $p < 0.001$; Supplementary Fig. 2). Only the distribution of $t$-values based on the actual, unshuffled KL-VS[het] labels was significantly greater than zero (inferior temporal ROI: $t(999) = −77.76$, $p < 0.001$; global ROI: $t(999) = −83.88$, $p < 0.001$) and the 95% confidence intervals did not include zero (inferior temporal ROI: 95% CI = [−4.847, −0.563]; global ROI: 95% CI = [−4.763, −0.794]). Together, these results confirmed a robust association between KL-VS[het] and lower Aβ-associated tau accumulation.

In order to determine whether our findings were driven by clinical diagnosis, we repeated the analyses in 156 MCI patients (34 KL-VS[het] carriers and 122 non-carriers) and found comparable KL-VS[het] × amyloid-PET interaction effects on tau-PET uptake (inferior temporal ROI: beta = −0.26, $p = 0.003$, $N = 156$, Cohen's $f = 0.251$; global ROI: beta = −0.25, $p = 0.004$, $N = 156$, Cohen's $f = 0.243$; Supplementary Fig. 1c, d). Repeating the analysis in 347 CN participants (102 KL-VS[het] carriers and 245 non-carriers) yielded no KL-VS[het] × amyloid-PET interaction effects on tau-PET levels in either ROI (both $p > 0.05$; Supplementary Fig. 1e, f).

A few participants showed lower tau-PET levels in the ROIs than in the reference region resulting in a tau-PET standard uptake value ratio (SUVR) < 1 (2 participants for the inferior temporal ROI and 51 participants for the global ROI). Yet, the results of the KL-VS[het] × amyloid interaction analyses on tau-PET levels remained significant after excluding those participants (Supplementary Fig. 1g, f).

**KL-VS heterozygosity is related to lower amyloid-dependent tau accumulation over time**. In a subsample of 200 participants in whom longitudinal tau-PET data were available, we investigated whether KL-VS^het attenuates the association between baseline amyloid-PET levels and the rate of change in tau-PET assessed over a time interval of 1.63 years on average (range: 1–4 years). We found a significant KL-VS^het × amyloid-PET interaction effect on tau-PET annual change rates in the inferior temporal ROI (beta = −0.22, $p$ = 0.039, N = 200, Cohen's $f$ = 0.148), but not in the global ROI (beta = −0.15, $p$ = 0.176, N = 200, Cohen's $f$ = 0.098). KL-VS^het carriers showed lower tau-PET increases in inferior temporal cortices over time as a function of rising global amyloid-PET levels (Fig. 1c, d) suggesting that the KL-VS^het variant might be protective against Aβ-associated increase of tau pathology. The main effects of amyloid-PET on tau-PET change rates for KL-VS^het carriers and non-carriers are reported in Supplementary Table 1.

**Stronger protective effect of KL-VS heterozygosity in ApoE ε4 carriers**. Previous studies have reported an ApoE ε4-genotype-dependent effect of KL-VS^het on amyloid-PET[16]. Hence, we additionally explored whether ApoE ε4 carriers showed a stronger association between KL-VS^het and lower tau accumulation than ApoE ε4 non-carriers, controlling for age, sex, education, diagnosis, and global amyloid-PET levels in the regression analyses. This analysis yielded a significant KL-VS^het × ApoE ε4 interaction effect on tau-PET levels (inferior temporal ROI: beta = −0.11, $p$ = 0.031, N = 551, Cohen's $f$ = 0.093; global ROI: beta = −0.10, $p$ = 0.041, N = 551, Cohen's $f$ = 0.088; Supplementary Fig. 3).

**Spatial match between KL mRNA expression and the effect of KL-VS heterozygosity on tau-PET**. In order to estimate the spatial overlap between the strength of $KL$ gene expression and the test statistic of the KL-VS^het × amyloid-PET interaction on tau-PET, we obtained whole-brain mRNA expression levels of $KL$ generated by post-mortem microarray assessments of six healthy brain donors and subsequently mapped to the Allen Brain Atlas[33,34]. We computed median scores of log2 mRNA expression of $KL$ across the six donors within 34 left-hemispheric regions of the Freesurfer-based Desikan–Killiany brain atlas[38]. We focused on the left hemisphere since all donors had microarray assessment available for the left hemisphere and only two donors had an assessment for the right hemisphere. Furthermore, we estimated the KL-VS^het × amyloid-PET interaction effect on tau-PET levels within the same 34 brain atlas regions using the aforementioned regression model. Surface mapping of both the KL-VS^het × amyloid-PET interaction effect (which were all in the same direction) and $KL$ mRNA expression is displayed in Fig. 2a–d. Spatial correlation analysis revealed a significant association ($r$ = 0.46, $p$ = 0.007; Fig. 2e). This result suggests that regions with higher $KL$ mRNA expression levels were more likely to display lower Aβ-related tau-PET levels in KL-VS^het carriers versus non-carriers. Visual inspection of the thresholded spatial maps indicated that those areas showing both a significant KL-VS^het × amyloid-PET interaction effect (Fig. 2b; see Supplementary Table 2 for detailed statistical results) and high $KL$ mRNA expression (log2 > 75th percentile) (Fig. 2d) were specifically located within the mesiotemporal and inferior and middle temporal brain regions and the posterior cingulum.

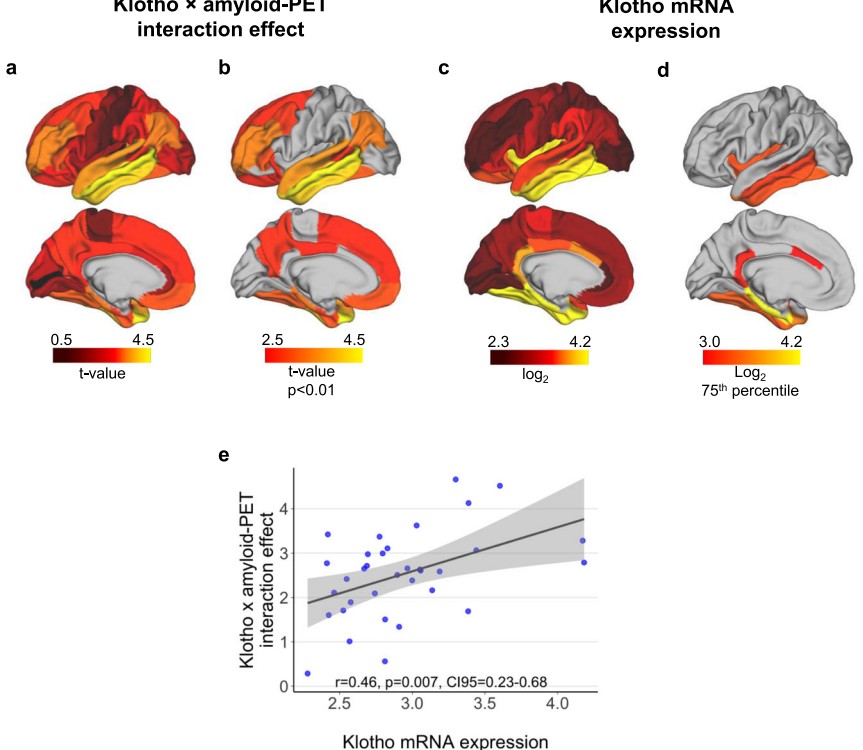

**Fig. 2 Spatial patterns of KL-VS^het-related attenuation of tau-PET and Klotho mRNA expression. a** Surface mapping of the interaction effect between KL-VS^het and amyloid-PET levels on tau-PET accumulation within 34 left-hemispheric regions of the Desikan–Killiany atlas. Yellow colors indicate higher $t$-values reflective of a stronger interaction effect (all $t$-values inverted for illustration purpose; see Supplementary Table 2 for details statistical results). **b** Thresholded spatial map color-code only regions with a significant ($p$ < 0.01, unadjusted for multiple comparisons) KL-VS^het × amyloid-PET interaction effect. **c** Surface mapping of median KL mRNA expression (i.e., log2 derived from the Allen Brain Atlas) within the identical 34 atlas regions. Yellow colors indicate higher $KL$ mRNA expression. **d** Thresholded spatial maps restricted to regions falling above the 75th percentile of $KL$ mRNA expression.
**e** Scatterplot depicting the association between ROI-based $KL$ mRNA expression and KL-VS^het × amyloid-PET interaction effect on regional tau-PET uptake. Statistical results are derived from the Pearson correlation (two-sided). Linear model fits are indicated together with 95% confidence intervals. Source data are provided as a Source Data file.

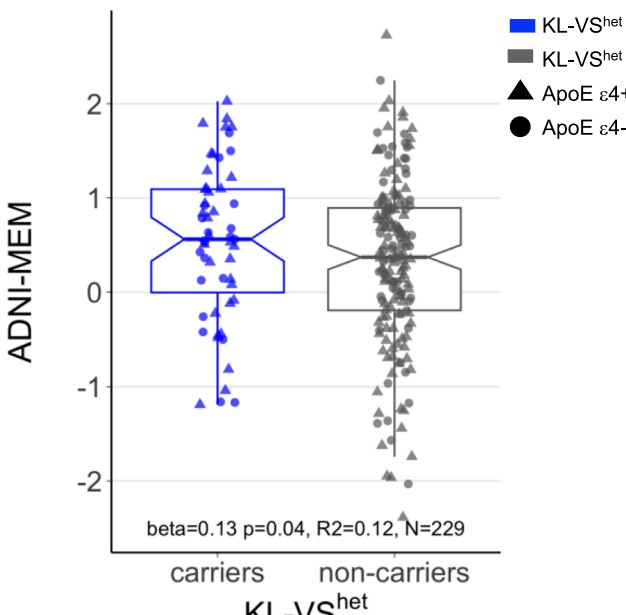

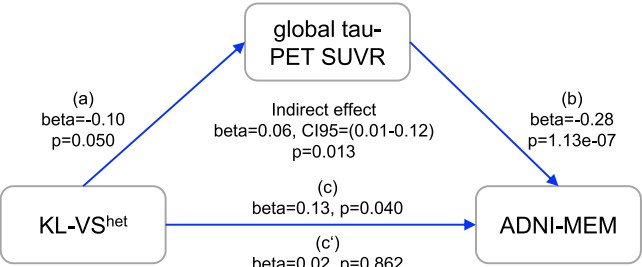

**Fig. 4 Lower tau-PET levels mediate the beneficial association of KL-VS heterozygosity and memory in individuals with the elevated amyloid-PET burden.** Path diagram of the mediation model (assessed only in amyloid-positive participants, $N = 229$), showing that the association between KL-VS$^{het}$ and better memory performance is mediated via lower global tau-PET uptake. Memory is measured by ADNI-MEM, i.e., an established memory composite score[39]. Path-weights are displayed as standardized beta values. All paths are controlled for age, sex, diagnosis, education, ApoE ε4 carrier status, and continuous global amyloid-PET levels. The significance of the indirect effect was determined using bootstrapping with 10,000 iterations as implemented in the mediation package in R.

**Fig. 3 Association between KL-VS heterozygosity and memory in amyloid-positive individuals.** Boxplot shows memory performance as a function of KL-VS$^{het}$ variant in individuals with a positive amyloid-PET (SUVR$_{FBP}$ ≥ 1.11 or SUVR$_{FBB}$ ≥ 1.08). Blue and gray colors indicate individuals with heterozygous ($N = 55$) or non-heterozygous KL-VS alleles ($N = 174$). Memory was measured by an established composite score, ADNI-MEM, based on test performance across multiple different memory tests[39]. Statistical result of the main effect of KL-VS$^{het}$ on memory was derived from multiple linear regression analysis, controlled for age, sex, diagnosis, education, and ApoE ε4 carrier status. Boxplots show the 25th percentile, median, 75th percentile (box), 95% confidence intervals of the median (notch), and 1.5× IQR (whiskers).

**Tau mediates the association between KL-VS heterozygosity and less memory impairment.** In the main analysis, we assessed whether KL-VS$^{het}$ is beneficial for memory functions via lowering tau pathology. Because the interaction effect of KL-VS$^{het}$ × amyloid-PET on tau-PET levels showed that KL-VS$^{het}$ is associated with lower tau accumulation at higher levels of amyloid-PET, we restricted our analysis to amyloid-positive participants. We used mediation analysis with 10,000 bootstrapping iterations in order to test whether KL-VS$^{het}$ is associated with better memory in individuals with elevated Aβ burden and whether this effect is mediated via reduced global tau-PET levels. Memory performance was measured by an established composite score based on participant's results across multiple different memory tests (ADNI-MEM)[39]. The mediation analysis was controlled for age, sex, education, diagnosis, ApoE ε4 status, and global amyloid-PET levels. Supporting our hypothesis, we found KL-VS$^{het}$ to be associated with higher ADNI-MEM scores (beta = 0.13, $p = 0.040$, Cohen's $f = 0.104$, $N = 229$, Fig. 3) and that this relationship was significantly mediated by lower tau-PET levels (bootstrapped average causal mediation effect: beta = 0.06, 95% CI = 0.01–0.12, $p = 0.013$, $N = 229$). This result indicates that, in individuals with an elevated Aβ burden, KL-VS$^{het}$ carriers showed less impaired episodic-memory abilities when compared to KL-VS$^{het}$non-carriers due to lower tau-PET levels in KL-VS$^{het}$ carriers. A path model of the mediation analysis is shown in Fig. 4.

In the secondary analysis, we addressed the question of whether KL-VS$^{het}$ exerts a beneficial influence on memory via lowered neurofibrillary tau in the whole sample without stratification based on amyloid-PET levels. To this end, we estimated the interaction effect between KL-VS$^{het}$ and amyloid-PET on ADNI-MEM scores,

with and without controlling for global tau-PET levels as a covariate. We reasoned that if the association between KL-VS$^{het}$ on cognition is mediated by tau-PET, the interaction between KL-VS$^{het}$ and amyloid-PET on cognition should be diminished when controlling for tau-PET. Without controlling tau-PET levels, we found a significant KL-VS$^{het}$ × amyloid-PET interaction effect on memory functions (beta = 0.08, $p = 0.037$, $N = 549$, Cohen's $f = 0.090$). Specifically, we observed that individuals with a high amyloid-PET burden showed better cognitive performance when being KL-VS$^{het}$ carriers compared to non-carriers (Supplementary Fig. 4). As hypothesized, the KL-VS$^{het}$ × amyloid-PET interaction on memory no longer reached significance level when global tau-PET levels were controlled for (beta = 0.06, $p = 0.125$, $N = 549$).

Besides our focus analysis of the beneficial effect of KL-VS$^{het}$ on memory, i.e., the cognitive domain affected early in AD, we also explored the effect on other cognitive domains including executive functions (composite score ADNI-EF), language (composite score ADNI-LAN) and visual-spatial perception (composite score ADNI-VS). In amyloid-positive participants, we found KL-VS$^{het}$ to be associated with higher ADNI-LAN scores (beta = 0.14, $p = 0.027$, Cohen's $f = 0.114$, $N = 229$; Supplementary Fig. 5a), a trend-level association with higher ADNI-EF scores (beta = 0.12, $p = 0.067$, Cohen's $f = 0.095$, $N = 229$) and no association with ADNI-VS scores (beta = 0.05, $p = 0.468$, $N = 229$). The beneficial association between KL-VS$^{het}$ and language abilities was mediated by lower global tau-PET levels in KL-VS$^{het}$ carriers versus non-carriers (bootstrapped average causal mediation effect: beta = 0.05, 95% CI = 0.01–0.12, $p = 0.017$, $N = 229$; Supplementary Fig. 5b).

**Is KL-VS heterozygosity associated with lower Aβ accumulation?** A recent study found a protective influence of KL-VS$^{het}$ on longitudinal amyloid-PET in cognitively unimpaired ApoE ε4 carriers aged between 60 and 80 years, but not in ApoE ε4 non-carriers or older participants[13]. In contrast to this earlier report, we did not find an age-dependent KL-VS$^{het}$ effect on cross-sectional amyloid-PET levels acquired at the time of tau-PET assessment in the current sample (KL-VS$^{het}$ × age interaction: beta = 0.21, $p = 0.597$, $N = 551$; Supplementary Fig. 6a) or an ApoE ε4-dependent KL-VS$^{het}$ effect in the subsample of CN participants aged between 60 and 80 years (beta = 0.04, $p = 0.586$, $N = 347$, Cohen's $f = 0.030$; Supplementary

Fig. 6b). However, more subtle effects may have been overlooked in the current more restricted sample of individuals undergoing both amyloid- and tau-PET. Therefore, in a supplementary analysis, we included all participants with amyloid-PET ($N = 1067$) from ADNI, regardless of whether or not they underwent tau-PET assessment. We found a trend-level significant KL-VS$^{het}$ × age interaction effect one amyloid-PET that demonstrated that KL-VS$^{het}$ carriers in the lower age range (<80 years) displayed lower amyloid-PET levels than non-carriers (beta = 0.53, $p = 0.046$, Cohen's $f = 0.061$, $N = 1067$; Supplementary Fig. 7a). Consistent with the earlier report, we found a significant KL-VS$^{het}$ × ApoE ε4 interaction effect on global amyloid-PET levels in CN participants aged between 60 and 80 years (beta = −0.121, $p = 0.043$, Cohen's $f = 0.095$, $N = 464$; Supplementary Fig. 7b). The same analysis in MCI participants within the same age range showed no significant interaction effect (beta = 0.02, $p = 0.780$, $N = 463$, Cohen's $f = 0.013$), suggesting that the association between KL-VS$^{het}$ and lower amyloid-PET uptake is restricted to a younger age and non-symptomatic cognitive status. See Supplementary Table 3 for detailed sample characteristics of the larger ADNI amyloid-PET sample compared to the current ADNI tau-PET sample. Thus, our results in the larger sample are consistent with those from Belloy et al.'s analysis on the effect of KL-VS$^{het}$ on amyloid-PET stratified by age and ApoE genotype while also showing that the effect size of KL-VS$^{het}$ on tau-PET is stronger than that on amyloid-PET.

## Discussion

The heterozygous *KL* gene variant KL-VS$^{het}$ has been previously associated with higher longevity and cognition performance in adulthood and reduced AD dementia risk[13]. We demonstrate that elderly KL-VS$^{het}$ carriers with elevated Aβ burden, i.e., the earliest primary AD pathology, exhibited lower tau-PET levels and tau-PET annual change rates when compared to those in KL-VS$^{het}$non-carriers. In amyloid-positive participants, the KL-VS$^{het}$ variant was associated with better memory performance, and this relationship was mediated by lower tau-PET levels, suggesting that lower levels of pathologic tau in the KL-VS$^{het}$ carriers explained the association between KL-VS$^{het}$ and better memory performance. Although our findings do not implicate a causative mechanism of Klotho in AD, we provide evidence for a potential protective role of KL-VS$^{het}$ against Aβ-dependent tau pathology that is the key AD brain alteration linked to cognitive impairment.

To our knowledge, the current study is the first to date that evaluated the interaction between KL-VS$^{het}$ and Aβ on tau accumulation and cognitive decline in humans. There is a growing literature on protective genetic variants in AD[40–42], but only a few studies have reported genetic variants to be associated with lower tau pathology in AD[43]. For the KL-VS$^{het}$ variant, previous studies reported an association with reduced Aβ accumulation in elderly ApoE ε4 risk-carriers[13,16]. We extend these previous findings by showing that the relationship between Aβ accumulation and fibrillar tau is modulated by KL-VS$^{het}$, such that lower local and global tau-PET levels were observed per unit increase of global amyloid-PET burden in KL-VS$^{het}$ carriers when compared to those in non-carriers. This is important because Aβ deposition precedes the development of dementia symptoms by up to 20 years[14], and as confirmed by a very recent longitudinal amyloid/tau-PET study, high baseline Aβ is associated with subsequent tau accumulation, while Aβ and tau in synergy lead to most pronounced subsequent cognitive decline[21]. The region showing one of the strongest interaction effects between KL-VS$^{het}$ and amyloid-PET on tau-PET was the inferior temporal gyrus (Fig. 2a), a brain area that typically shows an early Aβ-related increase in tau-PET[26] before elevated tau-PET levels extend to other higher cortical brain areas[20]. The protective association

between KL-VS$^{het}$ and tau-PET was present selectively in participants with abnormally elevated levels of amyloid-PET and more pronounced in ApoE ε4 carriers. Stratified analyses further revealed a significant KL-VS$^{het}$ effect in the MCI but not in the CN subgroup, which could potentially be due to a stage-dependent beneficial effect of Klotho. However, an alternative explanation is that the levels of both amyloid- and tau-PET are lower in CN compared to those in MCI, and thus any protective effect is likely to be of smaller size and would require a larger sample size to detect. Together, these results support the notion that KL-VS$^{het}$ is associated with an Aβ-related rather than age-related reduction of tau pathology.

In amyloid-positive individuals, we found KL-VS$^{het}$ to be associated with better memory performance, mediated by the effect of KL-VS$^{het}$ on tau-PET. Our results are broadly consistent with those from studies on healthy aging, reporting KL-VS$^{het}$ to be associated with better cognition[5,8–10], and lower risk of conversion from cognitively normal to mild cognitive impairment or AD dementia in ApoE ε4 carriers[13,16]. Our findings suggest that the association between KL-VS$^{het}$ and lower neurofibrillary tau pathology is of central importance for the association found between KL-VS$^{het}$ and less cognitive impairment. A previously reported absence of an association between KL-VS$^{het}$ and cognitive decline in asymptomatic participants with elevated levels of Aβ[44] did not assess the presence of abnormal neurofibrillary tau, which may have hampered to detect an effect of KL-VS$^{het}$ on cognitive decline in subjects at risk of AD[35].

Previous studies reported KL-VS$^{het}$ to be associated with lower amyloid-PET in ApoE ε4 carriers (but not in ApoE ε4 non-carriers), which was strongest in the age range between 60 and 80 years[13,16]. In our primary analysis, we did not confirm age- or ApoE ε4-dependent effects of KL-VS$^{het}$ on tau-PET. By investigating a larger sample of all participants with available amyloid-PET regardless of the availability of tau-PET ($N = 1067$), we were able to substantiate those earlier findings[13,16]. Specifically, we showed reduced amyloid-PET burden in younger KL-VS$^{het}$ carriers (<80 years) and, in accordance with previous work, this association was mainly driven by cognitively unimpaired ApoE ε4 carriers rather than non-carriers or MCI patients. Comparing effect sizes, Cohen's $f = 0.061$ for the association between KL-VS$^{het}$ and lower amyloid-PET versus $f = 0.114$ for the association with lower tau-PET, strengthens the important role of changes in tau pathology for understanding the role of Klotho in AD.

The mechanisms linking Klotho to tau pathology remain elusive. Klotho is a pleiotropic protein that has been implicated in multiple biological processes including insulin regulation[4], growth factor functions, in particular of FGF23[45], regulation of members of the redox system[46], and calcium signaling[47]. One possibility of how the Klotho protein might be linked to reduced neurofibrillary tau is its involvement in autophagy[48], a mechanism that is involved in the clearance of AD pathologies[49]. Lentiviral overexpression of Klotho protein in an APP-PS1 mouse model of Aβ deposition reduced Aβ plaque load in aged mice and rescued the impaired autophagy possibly by modulating the Akt/mTOR pathway[11,50]. Since APP-PS1 mice do not develop tau pathology, it remains, however, to be tested whether Klotho-induced autophagy reduces tau pathology. Those mechanistic explanations remain speculative at this point and the current work encourages future studies to investigate the mechanism that could underlie the protection Klotho exerts against the development of Aβ-related tau pathology.

Our findings of the spatial correspondence between the strength of the effect of KL-VS$^{het}$ on regional tau-PET and the spatial distribution of *KL* mRNA suggest a local effect of Klotho on the development of fibrillar tau, especially in temporal brain areas. Alternative splicing of the human *KL* mRNA results in both a membrane-bound and a

secreted transcript of Klotho[1,4], indicating that Klotho may act both in a cell-autonomous manner and as a humoral factor. Therefore, differences in gene expression in *KL* in the brain and/or different circulating levels of Klotho linked to KL-VS[het5] may influence the development of pathological tau[11], but this link remains to be investigated.

Our results have important implications for clinical trials in AD. Since tau pathology correlates more closely with clinical symptoms than Aβ, tau-targeted therapies seem a promising approach to arrest disease progression[51]. The common KL-VS genotype may inform those clinical trials that target tau pathology. Especially when anti-tau trials aim to include amyloid-positive or ApoE ε4 carrying participants, group differences in the KL-VS[het] variant may be taken into account when estimating the expected change in tau pathology over time, which would be useful in the computation of statistical power to detect a treatment effect. Furthermore, the current findings encourage future studies to test whether enhancing Klotho protein levels could reduce the development of tau pathology in amyloid-positive participants. The Klotho protein is druggable and could thus be made a target in the development of disease-modifying therapeutic approaches.

Several caveats should be considered when interpreting the current results. First, the human *KL* gene consists of three polymorphic variants. We decided to focus on the KL-VS haplotype given the existing evidence of its beneficial influence on Aβ and cognition in both mice and humans[5,8–10,13,52]. While the second variant C1818T (rs564481) is located on the fourth exon and likely has no functional consequences itself, the third variant G395A (rs1207568) is located in the promoter region and may be a potential regulatory site of KL. The two latter variants appear more frequent in Asian populations, where they have been linked to cardiovascular risk factors[53]. Related to the current research question, an investigation across three independent cohorts of oldest-old Danes found different polymorphic variants of *KL*, besides KL-VS, to be associated with better cognitive functions[7]. It has yet to be proven whether these other *KL* variants also support resilience in AD. Another caveat is that we did not measure Klotho protein levels in the serum or CSF. Circulating levels of Klotho decrease during aging[54] and are associated with cognitive performance[5] and gray matter volume[55] in cognitively unimpaired individuals. In patients with AD, CSF levels of Klotho are reduced[52], where the experimental reversal of reduced Klotho expression in transgenic mouse models exerted beneficial effects on Aβ and cognition[8,11,30]. While the KL-VS[het] variant has been associated with higher circulating levels of Klotho[55], it remains to be investigated whether the association between KL-VS[het] and pathological tau are mediated by higher protein levels in the CSF and brain tissue.

In summary, our findings revealed a protective association of KL-VS[het] on tau accumulation that particularly manifested in amyloid-positive individuals, where lower tau pathology was related to better cognitive functions. These findings may be particularly informative for clinical anti-tau trials[56] and may encourage future studies on enhancing Klotho protein levels as a therapeutic intervention to slow down the development of tau pathology and dementia in AD.

## Methods

**Sample characteristics.** A total of 551 participants were selected from ADNI phase 3 (ClinicalTrials.gov ID: NCT02854033) based on the availability of KL-VS and ApoE ε4 genotyping, T1-weighted MRI, [18]F-flortaucipir (FTP) tau-PET and [18]F-florbetapir (FBP) or [18]F-florbetaben (FBB) amyloid-PET. MR and PET imaging had to be acquired during the same study visit. In addition, a subsample of 200 participants with a follow-up tau-PET assessment was selected for the longitudinal analyses. The two single-nucleotide polymorphisms for KL-VS (rs9536314 for F352V, rs9527025 for C370S) and ApoE (rs429358, rs7412) were genotyped using DNA extracted by Cogenics from a 3 mL aliquot of EDTA blood. Participants were assigned to the heterozygous KL-VS group when they carried 1, but not 2, copies of the KL-VS haplotype. ApoE ε4 carriers were defined as individuals carrying at least one ε4 allele. Clinical classification was performed by the ADNI centers, dividing participants into cognitively normal (CN,

Mini-Mental State Examination [MMSE] > 24, CDR = 0, non-depressed), mild cognitively impairment (MCI; MMSE > 24, CDR = 0.5, objective memory-loss on the education adjusted Wechsler Memory) or AD dementia (ADD; 19 < MMSE < 24, CDR = 0.5–1.0, NINCDS/ADRDA criteria for probable AD are fulfilled). All participants provided written informed consent and all work complied with ethical regulations for work with human participants.

**MR and PET acquisition and preprocessing.** All imaging data were downloaded from the ADNI loni image archive (https://ida.loni.usc.edu).

Structural T1-weighted images were acquired on 3T scanners using a 3D MPRAGE sequence with 1 mm isotropic voxel-size and a TR = 2300 ms (detailed scan protocols can be found on https://adni.loni.usc.edu/wp-content/uploads/2017/07/ADNI3-MRI-protocols.pdf). Structural images were processed using Freesurfer (version 5.3.0) and parcellated according to the Desikan–Killiany atlas[57].

Tau-PET was assessed in 6 × 5 min blocks 75 min after intravenous bolus injection of [18]F-FTP. Amyloid-PET scans were obtained during 4 × 5 min time frames measured 50–70 min post injection of [18]F-FBP or 90–110 min post injection of [18]F-FBB. For both tau- and amyloid-PET we downloaded partially preprocessed data (http://adni.loni.usc.edu/methods/pet-analysis-method/pet-analysis/).

All PET images were coregistered to the corresponding T1-weighted image to make use of Freesurfer-derived masks in participants' high resolution, native space. SUVR scores were obtained by normalizing tau-PET images to the inferior cerebellar gray matter and amyloid-PET images to the whole cerebellum, following the previous recommendations[58]. In order to make FBP and FBB amyloid-PET measures comparable, we transformed SUVR scores into centiloid (CL) units using the established transformation formula (http://adni.loni.usc.edu/wp-content/themes/freshnews-dev-v2/documents/pet/ADNI Centiloids Final.pdf). For the analysis of longitudinal tau-PET, we additionally calculated annual tau-PET SUVR change rates as the difference between tau-PET SUVR scores measured at the follow-up versus baseline visit divided by the follow-up time in years.

**Tau- and amyloid-PET regions of interest.** For the analyses of tau-PET, we extracted mean SUVR scores from bilateral inferior temporal gyri marking Aβ-related increase of tau pathology to neocortical structures[20,26,35–37]. In addition, we assessed global tau-PET burden[59] as the size-weighted mean SUVR score across all Freesurfer regions, excluding hippocampus, thalamus, and basal ganglia due to commonly reported tracer off-target binding[60].

For the analysis of amyloid-PET images, we computed mean amyloid-PET levels from a global ROI spanning lateral and medial frontal, anterior and posterior cingulate, lateral parietal, and lateral temporal regions. Mean SUVR from these regions was also used for sample stratification into amyloid-positive participants based on established thresholds ($SUVR_{FBP} \geq 1.11$ or $SUVR_{FBB} \geq 1.08$; see "ADNI_UCBERKELEY_AV45_Methods_12.03.15.pdf" and "UCBerkeley_FBB_Methods_04.11.19.pdf" on the ADNI website).

**mRNA expression levels of *KL*.** Regional gene expression was obtained from publicly available microarray measurements of regional mRNA expression based on post-mortem data from the Allen Brain Atlas (http://human.brain-map.org). The Allen Brain atlas is based on more than 60,000 microarray probes collected from 3700 autopsy-based brain tissue samples from a total of six individuals aged 24–57 without a known history of neurological or psychiatric diseases[33,34]. Microarray-based log2 expression values of 20,737 genes within each of the 3700 samples were mapped back into MNI standard space by the Allen Brain Institute using stereotactic coordinates of the examined probes. The whole gene expression data have been recently mapped to the Freesurfer-based Desikan–Killiany atlas as median gene expression for probes falling within each of the 68 atlas ROIs[38]. Here, we specifically extracted median expression of *KL* mRNA within these Desikan–Killiany ROIs, to test a spatial correlation between *KL* expression and KL-VS[het] effects on local tau-PET uptake. Since microarray assessments and thus *KL* mRNA expression of all six Allen brain atlas subjects were available only for the left hemisphere (vs. two subjects for the right hemisphere), we restricted the analysis of *KL* mRNA expression data to the more robust estimates of the left hemisphere in line with previous studies[61,62].

**Neuropsychological assessment.** The ADNI neuropsychological test battery contains multiple indicators for memory functions, on which basis a composite score (ADNI-MEM) has been established[39]. ADNI-MEM summarizes test performance on the Rey Auditory Verbal Learning Test, elements from the AD Assessment Scale-Cognitive Subscale, word recall from the MMSE, and the Wechsler Logical Memory Scale II. In the exploratory analysis, we also used established ADNI summary scores of executive functions (ADNI-EF), language (ADNI-LAN), and visual-spatial abilities (ADNI-VS) (see ADNI_Methods_UWNPSYCHSUM_March_2020.pdf on ADNI webpage). Note that 2 participants had no neuropsychological tests available resulting in a sample of 549 participants for this part of the analysis.

**Statistical analysis.** All statistical analyses were conducted with R statistical software (version 3.6.1). *P* values were considered significant when meeting a two-tailed alpha threshold of 0.05. Baseline tau-PET SUVR values were entered as log-

transformed values into the statistical models to approximate normality. All interaction analyses were controlled for the main effect of the interaction terms. Group demographics were compared between KL-VS$^{het}$ carriers versus non-carriers using Welch $T$-tests for continuous measures and $\chi^2$ tests for categorical measures.

**KL-VS heterozygosity × amyloid interaction on tau pathology**. In our main analysis, we tested whether KL-VS$^{het}$ moderates the relationship between amyloid- and tau-PET. To this end, multiple linear regression analyses were used to estimate the interaction effects between KL-VS$^{het}$ and global amyloid-PET uptake on tau-PET levels in the inferior temporal ROI and the global ROI ($N = 551$). Age, sex, diagnosis, education, and ApoE ε4 carrier status were considered as covariates. In secondary analyses, we accounted for potential biases due to unequally sized KL-VS$^{het}$ groups by repeating the same interaction analysis in matched groups of equal size ($N = 288$). For this purpose, 144 out of 407 KL-VS$^{het}$non-carriers were selected based on propensity score matching for global amyloid-PET levels and clinical diagnosis using the matchit R package. To ensure that our results were not affected by the skewed distribution of PET data or outliers, we iteratively determined the t-statistic of the Kl-VS$^{het}$ × amyloid interaction effect on tau-PET levels using 1000 bootstrapping iterations (i.e., random sampling from the subject pool with replacement using the boot R package). As a reference, we generated a null distribution of the $t$-statistic using the same approach, but randomly reshuffling the KL-VS$^{het}$ labels on every iteration. We compared the mean $t$-value of the bootstrapped interaction effect to that of the null distribution using Welch $T$-tests. The significance of the bootstrapped interaction effect was determined by one-sample $t$-tests estimating whether the resulting distributions of t-values significantly differ from zero and by confirming that the 95% confidence intervals did not overlap with zero. In addition, we accounted for potential influences of clinical diagnoses by repeating the same interaction analyses in the subsample of only MCI ($N = 156$) or CN ($N = 347$) participants. Finally, we repeated the analysis in a subsample excluding participants with tau-PET SUVR values in the two ROIs smaller than the reference region (i.e., SUVR < 1).

Next, we tested whether KL-VS$^{het}$ moderates the relationship between amyloid- and tau-PET accumulation over time ($N = 200$). Separate linear regression models were used to estimate the KL-VS$^{het}$ × global amyloid-PET interaction effect on tau-PET annual change rates in the inferior temporal ROI and the global ROI, controlling for age, sex, and diagnosis.

**KL-VS heterozygosity × ApoE interaction on tau pathology**. Additional exploratory analyses were run to determine the influence of ApoE ε4 carrier status. To this end, we examined the interaction between KL-VS$^{het}$ and ApoE ε4 status on tau-PET levels in the whole sample (206 ApoE ε4 carriers and 345 non-carriers), controlling for age, sex, diagnosis, education, and global amyloid-PET uptake.

**Spatial match between *KL* mRNA expression and the association of KL-VS heterozygosity on tau pathology**. Next, we tested whether the favorable influence of KL-VS$^{het}$ on local tau-PET levels overlapped within those brain regions showing higher local *KL* mRNA expression levels. To this end, we determined *KL* mRNA expression using the Allen brain atlas data in all 34 left-hemispheric Desikan–Killiany atlas regions and determined the KL-VS$^{het}$ × amyloid-PET interaction effect on tau-PET uptake for corresponding anatomical regions. We then tested the ROI-to-ROI Pearson–moment correlation between regional *KL* mRNA expression and the interaction effect test statistic (not restricted to regions showing a significant interaction effect).

**KL-VS heterozygosity–memory relationship and lower tau pathology as a mediator**. We tested whether KL-VS$^{het}$ was associated with better memory functions, and whether this association was mediated by reduced tau-PET levels. For the main analysis, mediation analysis (causal mediation R package) was conducted in which KL-VS$^{het}$ variant was treated as a predictor, global tau-PET levels as a mediator, and ADNI-MEM scores as an outcome. Mediation analysis was performed in the subsample of amyloid-positive participants ($N = 229$) since we found KL-VS$^{het}$ to be associated with lower tau-PET levels specifically in individuals with elevated amyloid-PET uptake. Note that, since we conditioned the mediation effect on amyloid-PET levels, this is formally a moderated mediation analysis that we conducted only for one level of the moderator (amyloid status = positive) following our hypothesis. The significance of the mediation effect was determined using 10,000 bootstrapped iterations, where each path of the model was controlled for age, sex, diagnosis, education, ApoE ε4 carrier status, and global amyloid-PET levels.

We ran an alternative analysis strategy in the whole sample (including amyloid-positive and -negative participants, $N = 549$) that estimated the KL-VS$^{het}$ × amyloid-PET interaction effect on ADNI-MEM scores. Importantly, the interaction analysis was once run without and once with controlling global tau-PET levels. We specifically hypothesized that if the beneficial influence of KL-VS$^{het}$ is dependent on lowering tau accumulation, then the interaction effect should be diminished in the tau-controlled analysis. Other covariates considered in the multiple regression models were age, sex, diagnosis, education, and ApoE ε4 carrier status. In secondary analyses, we repeated the mediation analysis for ADNI summary scores of other cognitive domains.

**Is KL-VS heterozygosity associated with lower Aβ accumulation?** Lastly, we performed an exploratory analysis with the aim to confirm previously observed age- and ApoE-dependent associations between KL-VS$^{het}$ and lower amyloid-PET burden[13]. For this purpose, we tested for a KL-VS$^{het}$ × age effect on global amyloid-PET levels in the current sample ($N = 551$) and in a larger ADNI sample ($N = 1067$) including all participants with amyloid-PET assessment and KL-VS status (regardless of whether or not they underwent tau-PET assessment). Sex, education, and diagnosis were considered as covariates. Comparable to the original report, we also investigated ApoE-dependent effects of KL-VS$^{het}$ on amyloid-PET levels in a subgroup including only CN participants aged between 60 and 80 years. Age, sex, and education were considered as covariates.

**Reporting summary**. Further information on research design is available in the Nature Research Reporting Summary linked to this article.

## Data availability
The data that support the findings of this study were obtained from the Alzheimer's Disease Neuroimaging Initiative (ADNI) and are available from the ADNI database (adni.loni.usc.edu) upon registration and compliance with the data use agreement. A list including the anonymized participant identifiers of the currently used sample and the source file can be downloaded from the ADNI database (tau-PET data release in May 2020; UCBERKELEYAV1451_05_12_20.csv). The Allen Brain Atlas (http://human.brain-map.org) and Freesurfer-mapped transcriptomic data from the Allen Brain Atlas (http://figshare.com/articles/A_FreeSurfer_view_of_the_cortical_transcriptome_generated_from_the_Allen_Human_Brain_Atlas/1439749) are freely available online. Source data underlying Fig. 2 are provided with this paper.

## Code availability
The R code pertaining to the figures in this manuscript is provided at https://github.com/njulianeitzel/NatCommun2021_KL-VS. Costume R code can be obtained from the first author upon request.

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

## Acknowledgements

Data used in the preparation of this manuscript were obtained from the ADNI database (adni. loni.usc.edu). As such, the investigators within the ADNI study contributed to the design and implementation of ADNI and/or provided data but did not participate in the analysis or writing of this paper. The study was funded by DAAD post-doc fellowship (to J. N.), grants from the Alzheimer Forschung Initiative (AFI, Grant 15035 to M. E.), Legerlotz Stiftung (to M. E.), LMUexcellent (to M. E.), Deutsche Forschungsgemeinschaft (DFG, German Research Foundation) grant for major research instrumentation (DFG, INST 409/193-1 FUGG; to M. D.); Hertie Foundation for Clinical Neurosciences (to N. F.), LMU Förderung Forschung Lehre (Reg. 1032 to N. F.), European Union's Horizon 2020 research and innovation programme (grant agreement No. 666881 [SVDs@target] and 667375 [CoSTREAM]; to M. D.), the DFG as part of the Munich Cluster for Systems Neurology (EXC 2145 SyNergy—ID 390857198) and the CRC 1123 (B3) to M. D.). M. B. received speaker honoraria from GE healthcare and LMI and is an advisor of LMI. ADNI data collection and sharing for this project was funded by the ADNI (National Institutes of Health Grant U01 AG024904) and DOD ADNI (Department of Defense award number W81XWH-12-2-0012). ADNI is funded by the National Institute on Aging, the National Institute of Biomedical Imaging, and Bioengineering, and through contributions from the following: AbbVie, Alzheimer's Association; Alzheimer's Drug Discovery Foundation; Araclon Biotech; BioClinica, Inc.; Biogen; Bristol-Myers Squibb Company; CereSpir, Inc.; Cogstate; Eisai Inc.; Elan Pharmaceuticals, Inc.; Eli Lilly and Company; EuroImmun; F. Hoffmann-La Roche Ltd and its affiliated company Genentech, Inc.; Fujirebio; GE Healthcare; IXICO Ltd.; Janssen Alzheimer Immunotherapy Research & Development, LLC.; Johnson & Johnson Pharmaceutical Research & Development LLC.; Lumosity; Lundbeck; Merck & Co., Inc.; Meso Scale Diagnostics, LLC.; NeuroRx Research; Neurotrack Technologies; Novartis Pharmaceuticals Corporation; Pfizer Inc.; Piramal Imaging; Servier; Takeda Pharmaceutical Company; and Transition Therapeutics. The Canadian Institutes of Health Research is providing funds to support ADNI clinical sites in Canada. Private sector contributions are facilitated by the Foundation for the National Institutes of Health (www.fnih.org).

## Author contributions

J.N.: study concept and design, data processing, statistical analysis, interpretation of the results, and writing the manuscript. N.F.: critical revision of the manuscript. A.R.: data processing and critical revision of the manuscript. M.D.: critical revision of the manuscript. M.B.: critical revision of the manuscript. R.M.: data processing and critical revision of the manuscript. M.E.: study concept and design, interpretation of the results, and writing the manuscript. ADNI provided all data used for this study.

## Funding

## Competing interests

M.B. received speaker honoraria from GE healthcare and LMI and is an advisor of LMI. All other authors declare no competing interests.

## Additional information

## Alzheimer's Disease Neuroimaging Initiative (ADNI)

Michael Weiner[7], Paul Aisen[8], Ronald Petersen[9], Clifford R. Jack Jr.[9], William Jagust[10], John Q. Trojanowki[11], Arthur W. Toga[12], Laurel Beckett[13], Robert C. Green[14], Andrew J. Saykin[15], John Morris[16], Leslie M. Shaw[17], Enchi Liu[18], Tom Montine[19], Ronald G. Thomas[8], Michael Donohue[8], Sarah Walter[8], Devon Gessert[8], Tamie Sather[8], Gus Jiminez[8], Danielle Harvey[13], Matthew Bernstein[9], Nick Fox[20], Paul Thompson[21], Norbert Schuff[22], Charles DeCArli[13], Bret Borowski[9], Jeff Gunter[9], Matt Senjem[9], Prashanthi Vemuri[9], David Jones[9], Kejal Kantarci[9], Chad Ward[9], Robert A. Koeppe[23], Norm Foster[24], Eric M. Reiman[25], Kewei Chen[25], Chet Mathis[26], Susan Landau[10], Nigel J. Cairns[16], Erin Householder[16], Lisa Taylor Reinwald[16], Virginia Lee[27], Magdalena Korecka[27], Michal Figurski[27], Karen Crawford[12], Scott Neu[12], Tatiana M. Foroud[15], Steven Potkin[28], Li Shen[15], Faber Kelley[15], Sungeun Kim[15], Kwangsik Nho[15], Zaven Kachaturian[29], Richard Frank[30], Peter J. Snyder[31], Susan Molchan[32], Jeffrey Kaye[33], Joseph Quinn[33], Betty Lind[33], Raina Carter[33], Sara Dolen[33], Lon S. Schneider[34], Sonia Pawluczyk[34], Mauricio Beccera[34], Liberty Teodoro[34], Bryan M. Spann[34], James Brewer[35], Helen Vanderswag[35], Adam Fleisher[25], Judith L. Heidebrink[23], Joanne L. Lord[23], Sara S. Mason[9], Colleen S. Albers[9], David Knopman[9], Kris Johnson[9], Rachelle S. Doody[36], Javier Villanueva Meyer[36], Munir Chowdhury[36], Susan Rountree[36], Mimi Dang[36], Yaakov Stern[37], Lawrence S. Honig[37], Karen L. Bell[37], Beau Ances[38], John C. Morris[38], Maria Carroll[38], Sue Leon[38], Mark A. Mintun[38], Stacy Schneider[38], Angela OliverNG[39], Randall Griffith[39], David Clark[39], David Geldmacher[39], John Brockington[39], Erik Roberson[39], Hillel Grossman[40], Effie Mitsis[40], Leyla deToledo-Morrell[41], Raj C. Shah[41], Ranjan Duara[42], Daniel Varon[42], Maria T. Greig[42], Peggy Roberts[42], Marilyn Albert[43], Chiadi Onyike[43], Daniel D'Agostino II[43], Stephanie Kielb[43], James E. Galvin[44], Dana M. Pogorelec[44], Brittany Cerbone[44], Christina A. Michel[44], Henry Rusinek[44], Mony J. de Leon[44], Lidia Glodzik[44], Susan De Santi[44], P. Murali Doraiswamy[45], Jeffrey R. Petrella[45], Terence Z. Wong[45], Steven E. Arnold[17], Jason H. Karlawish[17], David Wolk[17], Charles D. Smith[46], Greg Jicha[46], Peter Hardy[46], Partha Sinha[46], Elizabeth Oates[46], Gary Conrad[46], Oscar L. Lopez[26], MaryAnn Oakley[26], Donna M. Simpson[26],

Anton P. Porsteinsson[47], Bonnie S. Goldstein[47], Kim Martin[47], Kelly M. Makino[47], M. Saleem Ismail[47], Connie Brand[47], Ruth A. Mulnard[48], Gaby Thai[48], Catherine Mc Adams Ortiz[48], Kyle Womack[49], Dana Mathews[49], Mary Quiceno[49], Ramon Diaz Arrastia[49], Richard King[49], Myron Weiner[49], Kristen Martin Cook[49], Michael DeVous[49], Allan I. Levey[50], James J. Lah[50], Janet S. Cellar[50], Jeffrey M. Burns[51], Heather S. Anderson[51], Russell H. Swerdlow[51], Liana Apostolova[52], Kathleen Tingus[52], Ellen Woo[52], Daniel H. S. Silverman[52], Po H. Lu[52], George Bartzokis[52], Neill R. Graff Radford[53], Francine ParfittH[53], Tracy Kendall[53], Heather Johnson[53], Martin R. Farlow[15], Ann Marie Hake[15], Brandy R. Matthews[15], Scott Herring[15], Cynthia Hunt[15], Christopher H. van Dyck[54], Richard E. Carson[54], Martha G. MacAvoy[54], Howard Chertkow[55], Howard Bergman[55], Chris Hosein[55], Sandra Black[56], Bojana Stefanovic[56], Curtis Caldwell[56], Ging Yuek Robin Hsiung[57], Howard Feldman[57], Benita Mudge[57], Michele Assaly Past[57], Andrew Kertesz[58], John Rogers[58], Dick Trost[58], Charles Bernick[59], Donna Munic[59], Diana Kerwin[60], Marek Marsel Mesulam[60], Kristine Lipowski[60], Chuang Kuo Wu[60], Nancy Johnson[60], Carl Sadowsky[61], Walter Martinez[61], Teresa Villena[61], Raymond Scott Turner[62], Kathleen Johnson[62], Brigid Reynolds[62], Reisa A. Sperling[63], Keith A. Johnson[63], Gad Marshall[63], Meghan Frey[63], Jerome Yesavage[64], Joy L. Taylor[64], Barton Lane[64], Allyson Rosen[64], Jared Tinklenberg[64], Marwan N. Sabbagh[65], Christine M. Belden[65], Sandra A. Jacobson[65], Sherye A. Sirrel[65], Neil Kowall[66], Ronald Killiany[66], Andrew E. Budson[66], Alexander Norbash[66], Patricia Lynn Johnson[66], Thomas O. Obisesan[67], Saba Wolday[67], Joanne Allard[67], Alan Lerner[68], Paula Ogrocki[68], Leon Hudson[68], Evan Fletcher[69], Owen Carmichael[69], John Olichney[69], Charles DeCarli[69], Smita Kittur[70], Michael Borrie[71], T. Y. Lee[71], Rob Bartha[71], Sterling Johnson[72], Sanjay Asthana[72], Cynthia M. Carlsson[72], Steven G. Potkin[73], Adrian Preda[73], Dana Nguyen[73], Pierre Tariot[25], Stephanie Reeder[25], Vernice Bates[74], Horacio Capote[74], Michelle Rainka[74], Douglas W. Scharre[75], Maria Kataki[75], Anahita Adeli[75], Earl A. Zimmerman[76], Dzintra Celmins[76], Alice D. Brown[76], Godfrey D. Pearlson[77], Karen Blank[77], Karen Anderson[77], Robert B. Santulli[78], Tamar J. Kitzmiller[78], Eben S. Schwartz[78], Kaycee M. SinkS[79], Jeff D. Williamson[79], Pradeep Garg[79], Franklin Watkins[79], Brian R. Ott[80], Henry Querfurth[80], Geoffrey Tremont[80], Stephen Salloway[81], Paul Malloy[81], Stephen Correia[81], Howard J. Rosen[7], Bruce L. Miller[7], Jacobo Mintzer[82], Kenneth Spicer[82], David Bachman[82], Elizabether Finger[83], Stephen Pasternak[83], Irina Rachinsky[83], Dick Drost[83], Nunzio Pomara[84], Raymundo Hernando[84], Antero Sarrael[84], Susan K. Schultz[85], Laura L. Boles Ponto[85], Hyungsub Shim[85], Karen Elizabeth Smith[85], Norman Relkin[86], Gloria Chaing[86], Lisa Raudin[86], Amanda Smith[87], Kristin Fargher[87] & Balebail Ashok Raj[87]

[7]UC San Francisco, San Francisco, CA, USA. [8]UC San Diego, San Diego, CA, USA. [9]Mayo Clinic, Rochester, NY, USA. [10]UC Berkeley, Berkeley, CA, USA. [11]University of Pennsylvania, Pennsylvania, CA, USA. [12]USC, Los Angeles, CA, USA. [13]UC Davis, Davis, CA, USA. [14]Brigham and Women's Hospital, Harvard Medical School, Boston, MA, USA. [15]Indiana University, Bloomington, IN, USA. [16]Washington University St. Louis, St. Louis, MO, USA. [17]University of Pennsylvania, Philadelphia, PA, USA. [18]Janssen Alzheimer Immunotherapy, South San Francisco, CA, USA. [19]University of Washington, Seattle, WA, USA. [20]University of London, London, UK. [21]USC School of Medicine, Los Angeles, CA, USA. [22]UCSF MRI, San Francisco, CA, USA. [23]University of Michigan, Ann Arbor, MI, USA. [24]University of Utah, Salt Lake City, UT, USA. [25]Banner Alzheimer's Institute, Phoenix, AZ, USA. [26]University of Pittsburgh, Pittsburgh, PA, USA. [27]UPenn School of Medicine, Philadelphia, PA, USA. [28]UC Irvine, Newport Beach, CA, USA. [29]Khachaturian, Radebaugh & Associates Inc and Alzheimer's Association's Ronald and Nancy Reagan's Research Institute, Chicago, IL, USA. [30]General Electric, Boston, MA, USA. [31]Brown University, Providence, RI, USA. [32]National Institute on Aging/National Institutes of Health, Bethesda, MD, USA. [33]Oregon Health and Science University, Portland, OR, USA. [34]University of Southern California, Los Angeles, CA, USA. [35]University of California San Diego, San Diego, CA, USA. [36]Baylor College of Medicine, Houston, TX, USA. [37]Columbia University Medical Center, New York, NY, USA. [38]Washington University, St. Louis, MO, USA. [39]University of Alabama Birmingham, Birmingham, MO, USA. [40]Mount Sinai School of Medicine, New York, NY, USA. [41]Rush University Medical Center, Chicago, IL, USA. [42]Wien Center, Vienna, Austria. [43]Johns Hopkins University, Baltimore, MD, USA. [44]New York University, New York, NY, USA. [45]Duke University Medical Center, Durham, NC, USA. [46]University of Kentucky, Lexington, NC, USA. [47]University of Rochester Medical Center, Rochester, NY, USA. [48]University of California, Irvine, CA, USA. [49]University of Texas Southwestern Medical School, Dallas, TX, USA. [50]Emory University, Atlanta, GA, USA. [51]University of Kansas, Medical Center, Lawrence, KS, USA. [52]University of California, Los Angeles, CA, USA. [53]Mayo Clinic, Jacksonville, FL, USA. [54]Yale University School of Medicine, New Haven, CT, USA. [55]McGill University, Montreal Jewish General Hospital, Montreal, WI, USA. [56]Sunnybrook Health Sciences, Toronto, ON, Canada. [57]U.B.C. Clinic for AD & Related Disorders, British Columbia, BC, Canada. [58]Cognitive Neurology St. Joseph's, Toronto, ON, Canada. [59]Cleveland Clinic Lou Ruvo Center for Brain Health, Las Vegas, NV, USA. [60]Northwestern University, Evanston, IL, USA. [61]Premiere Research Inst Palm Beach Neurology, West Palm Beach, FL, USA. [62]Georgetown University Medical Center, Washington, DC, USA. [63]Brigham and

Women's Hospital, Boston, MA, USA. [64]Stanford University, Santa Clara County, CA, USA. [65]Banner Sun Health Research Institute, Sun City, AZ, USA. [66]Boston University, Boston, MA, USA. [67]Howard University, Washington, DC, USA. [68]Case Western Reserve University, Cleveland, OH, USA. [69]University of California, Davis Sacramento, CA, USA. [70]Neurological Care of CNY, New York, NY, USA. [71]Parkwood Hospital, Parkwood, CA, USA. [72]University of Wisconsin, Madison, WI, USA. [73]University of California, Irvine BIC, Irvine, CA, USA. [74]Dent Neurologic Institute, Amherst, MA, USA. [75]Ohio State University, Columbus, OH, USA. [76]Albany Medical College, Albany, NY, USA. [77]Hartford Hospital, Olin Neuropsychiatry Research Center, Hartford, CT, USA. [78]Dartmouth Hitchcock Medical Center, Albany, NY, USA. [79]Wake Forest University Health Sciences, Winston-Salem, NC, USA. [80]Rhode Island Hospital, Providence, RI, USA. [81]Butler Hospital, Providence, RI, USA. [82]Medical University South Carolina, Charleston, SC, USA. [83]St. Joseph's Health Care, Toronto, ON, Canada. [84]Nathan Kline Institute, Orangeburg, SC, USA. [85]University of Iowa College of Medicine, Iowa City, IA, USA. [86]Cornell University, Ithaca, NY, USA. [87]University of South Florida, USF Health Byrd Alzheimer's Institute, Tampa, FL, USA.

