## [Peer Review File · Nature Communications]

REVIEWER COMMENTS

Reviewer #1 (Remarks to the Author):

Julia et al. investigated the association of KL-VS gene, which has been reported having predictive effect in Alzheimer's disease (AD), amyloid, tau and cognition on the AD continuum. They found KL-VS carriers had lower tau PET value than KL-VS non-carriers in individuals with abnormal amyloid PET, and KL-VS was related to better memory functions, which was mediated by lower tau PET. These findings provide new insight of the role of KL-VS gene in progression of AD, and how KL-VS affects amyloid and tau pathologies on the AD continuum, which may be very useful for prevention of AD, particularly for anti-tau clinical trials. Overall, I think this is a very important paper. The manuscript was very well written, and easily to be followed for me. I have a few comments as follows:

- 1) Please confirm the amyloid tracer, it seems like you used florbetapir tracer (PET signal 50-70 min post-injection) for amyloid PET, but you wrote florbetaben (PET signal 90-110min post-injection) in 341 line. By the way, you can also include florbetaben amyloid PET for cross-sectional data, which may increase the sample size for this study.
- 2) It may be better to define the cutoff of Global amyloid PET SUVR by using the whole ADNI sample rather than using only those subjects included in this study. For comparison, the cutoff 1.11 of AV45 suggested by ADNI PET core group may be also used.
- 3) Did you check the normality of the data in this study? If so, please explain the approach for normality test in "Statistical analysis".
- 4) In line 397, "To account for potential influences of clinical diagnoses, we ran the same analyses in the subsample of only MCI patients." What about CN group? It may be informative to include the results of CN group in supplemental material.
- 5) The authors only analyzed cross-sectional tau PET. What about doing the analyses for longitudinal tau changes in amyloid positive individuals or the whole sample (the sample size may be limited for amyloid positive individuals only). As far as I know, at least 185 ADNI participants had baseline amyloid PET, tau PET and longitudinal tau PET within one year according to one latest ADNI study (Guo et al. Biological Psychiatry, 2020, 10.1016/j.biopsych.2020.06.029). This study supports that amyloid pathology drives tau accumulation on the AD continuum, which may be useful for "Introduction" and "Discussion" of this paper as well.
- 6) Regarding to the cognition, it may be informative to include other domains of cognitive score provide in ADNI, such as, executive function, language and visuospatial function. As far as I know, they are all available as well as memory. So, it would be very easy to do the analyses of KL-VS, tau and cognition in line 168.
- 7) Since the authors only observed predictive effect of tau aggregation in amyloid positive individuals, can you comment why no significant effect was found in the whole sample or amyloid negative individuals?

Reviewer #2 (Remarks to the Author):

Summary:

The authors investigated the role of Klotho-VS heterozygosity (KL-VShet) carriership within the context of Alzheimer's disease. While previous work already showed that carriership was associated with reduced levels of amyloid deposition in APOE- ϵ 4 carriers, the current paper focused on downstream effects, namely tau accumulation and cognitive performance. The authors show that in elderly subjects with elevated amyloid burden, KL-VShet carriership was associated to lower tau burden as measured with PET imaging. In addition, there seemed to be a local association, as the spatial analyses revealed that Klotho was most strongly associated with temporal tau burden. Finally, KL-VShet carriership was associated with better memory performance, which was mediated by tau-PET levels, further supporting the influence of this gene on tau pathology.

The paper is novel, includes a respectable sample size, and is complete in terms of statistical analyses. In general, the paper is well written, but the methods section requires some clarification.

Major comments:

The methods section, specifically the statistical analyses subsection, is sometimes unclear and confusing. I suggest thorough rewriting and maybe the use of subheadings. Specific comments are:

- Sample, page 14, line 325-328: Please add the N of each included Dx subgroup.
- MR and PET acquisition and preprocessing, page 15, line 341: The authors state '50-70 min post-injection of 18F-AV45 Florbetaben'. There are several possible mistakes here. Was the tracer used Florbetaben? Then it should not state AV45 (which is Florbetapir) and the acquisition interval should be 90-110 p.i.. If indeed Florbetaben was used, then I hope that the correct interval (i.e. 90-110 p.i.) was used to obtain the SUVR data, as in accordance with the ADNI processing and product manufacturer guidelines. Please check and correct.
- Statistical analyses, page 17/18: The analyses described are jumping back and forth regarding the selection and use of subsamples. For example, in line 397 it is stated that the analyses was rerun only in MCI patients. Subsequently the next sentence starts with 'Additional'; was this analyses then done in the whole sample, or in the previously described MCI subsample? Please improve clarity. Also, in line 399-403 first a stratified analyses is described followed by the analyses in the whole cohort. This is also a bit of a confusing order to list step-wise analyses.
- Statistical analyses, page 18, lines 418-420: It is unclear whether only subjects with A β + status were thus included for this sub-analyses. Also, please specify how the optimal threshold was determined in addition to referencing the supplementary material as this is quite key in an early population.
- Statistical analyses, page 18, line 423: was the interaction analyses presented here with continuous amyloid-PET or with A β status?
- Throughout: add the N when any subgroup is mentioned.

Additional comments:

- Results, page 5, line 120: the finding that the KL-VShet was protective against A β -associated accumulation of tau pathology was also repeated in the MCI cohort. Considering that the tau-PET ROI used in this work was aimed to be sensitive of early pathology, it would be interesting to repeat the analyses also in the CN population. The sample size of this diagnostic group would allow for such stratification.
- Discussion, page 12, line 278-286: the spatial correspondence is discussed in this paragraph of the discussion section. However, I feel that one important aspect is missing and should be considered as an explanation; considering the relatively early population used in this work (which is an asset!), the finding of an association between Klotho and tau pathology in temporal brain areas, could be due to a power issue, as more wide-spread cortical tau is probably less frequently observed in this population. Please add this as a consideration when interpreting these results.
- Discussion, page 13, line 306-307: the sentence 'In summary, due to the high clinical relevance of tau accumulation and mixed results from trials on A β immunization, strategies for targeting tau pathology are moving into focus.' is a weird sentence to start the conclusion with, especially considering that after the words 'In summary' you don't summarize your own results. Also, the utility of this work in the context of trials is not discussed at all, which I would opt for including in the paper.

Reviewer #3 (Remarks to the Author):

In this study, Neitzel and colleagues sought to determine whether KL-VShet is associated with lower levels of pathologic tau and whether KL-VShet was also associated with lower memory impairment. For this purpose, they studied 354 individuals including CN, MCI, and AD; of whom 92 were KL-VShet and 262 were KL-VS noncarriers. They found that KL-VShet showed lower cross-sectional increase in tau-PET per unit increase in amyloid-PET when compared to non-carriers. This effect of KL-VShet on tau-PET also seemed stronger in Klotho mRNA-expressing brain regions mapped onto a gene expression atlas. Lastly, KL-VShet was related to better memory functions and this association was mediated by lower tau-PET.

This is overall a well-written paper. Some comments for the authors' consideration follow:

1. As specified in the title and other sections, the paper's purported objective is to test whether KL-

VShet is associated with reduced tau. This however is not what was actually done. They authors instead examined whether KL-VShet modifies the association between abeta and tau. Recommend either re-aligning analyses with purported aim or rewording the title (and all other affected sections) to align with what was actually done

2. The analyses, as stated above, are premised on KL-VShet modifying the association between abeta and tau. However, the authors reported (Table 1) that the KL-VS groups do not differ on abeta levels. I believe this has considerable implications for the rest of the models

3. The PET data appear to include several cases where SUVR is less than 1. Such data are functionally meaningless, and I suggest removing them from the analyses

4. Similarly, to describe "elevated amyloid burden" as an SUVR of >1 is neither acceptable nor in line with the extensive body of work in this area. If the authors would like to retain this analysis in the paper, they are encouraged to recalibrate their definition of elevated abeta.

5. Given the structure of the paper (that the findings from the tau model tie back into the memory model), it's concerning that the tau model and the memory model are not fit in concordance with each other. As mentioned above, the tau model is about KL-VShet modifying the association between abeta and tau whereas the memory model is about KL-VShet directly leading to better memory scores

6. It is customary when reporting a significant interaction (eg of KL-VS and abeta on tau) to follow it up by reporting on the simple slopes (eg the association of abeta and tau in the two KL-VS groups). This was not done in this case

7. The sample sizes in the KL-VShet vs KL-VS noncarrier groups are markedly disproportionate. This raises the possibility that the lack of association in the KL-VShet group is driven by the smaller sample. It would be important to reassure readers that this was not the case. One approach to doing this would be to use a matching algorithm to identify 92 KL-VS noncarriers who are matched to the 92 KL-VShet on relevant covariates; then repeat the analyses in this matched sample

8. The inclusion of all three diagnostic groups in the analyses exposes the models and conclusion to muddling. Suggest either focusing on only one diagnostic group or running the analyses separately in each of the diagnostic groups

9. The purpose and significance of the mRNA analyses was unclear to me. Importantly, p of .05 is inappropriate for whole brain analyses, even under exploratory conditions

Reviewer #1 (Remarks to the Author):

Julia et al. investigated the association of KL-VS gene, which has been reported having predictive effect in Alzheimer's disease (AD), amyloid, tau and cognition on the AD continuum. They found KL-VS carriers had lower tau PET value than KL-VS non-carriers in individuals with abnormal amyloid PET, and KL-VS was related to better memory functions, which was mediated by lower tau PET. These findings provide new insight of the role of KL-VS gene in progression of AD, and how KL-VS affects amyloid and tau pathologies on the AD continuum, which may be very useful for prevention of AD, particularly for anti-tau clinical trials. Overall, I think this is a very important paper. The manuscript was very well written, and easily to be followed for me. I have a few comments as follows:

1) Please confirm the amyloid tracer, it seems like you used florbetapir tracer (PET signal 50-70 min post-injection) for amyloid PET, but you wrote florbetaben (PET signal 90-110min post-injection) in 341 line. By the way, you can also include florbetaben amyloid PET for cross-sectional data, which may increase the sample size for this study.

Response: We thank the reviewer for the careful reading. We did use the florbetapir tracer previously, and now added also florbetaben (FBB) PET scans as suggested by the reviewer. In order to pool data across both tracers, we transformed the SUVR data into centiloid units as established in ADNI by Jagust and Landau http://adni.loni.usc.edu/wp-content/themes/freshnews-dev-v2/documents/pet/ADNI_Centiloids_Final.pdf. This increased the sample size of subjects with both cross-sectionally assessed tau-PET sample and any amyloid-PET substantially from 354 to 551 participants. We rerun all analyses in the updated larger sample. The results in this now larger sample were consistent with those reported in the submitted version of the manuscript. Briefly, we confirmed a significant amyloid x KL-VS^{het} interaction effect on tau-PET levels in the inferior temporal (beta = -0.12, p = 0.009, N = 551, Cohen's f = 0.112; **Fig. 1a**) and global ROI (beta = -0.13, p = 0.008, N=551, Cohen's f = 0.114; **Fig. 1b**) suggesting that the KL-VS^{het} variant attenuates the association between amyloid and tau pathology. The beneficial effect of KL-VS^{het} remained when testing only the subgroup of MCI patients (inferior temporal ROI: beta = -0.26, p = 0.003, N = 156, Cohen's f = 0.251; global ROI: beta = -0.25, p = 0.004, N = 156, Cohen's f = 0.243; **Supplementary Fig. 1c, d**). We found a significant spatial overlap between the strength of KL gene expression and the test statistic of the KL-VS^{het} x amyloid-PET interaction on tau-PET (r = 0.46, p = 0.007; **Figure 2e**). The updated description of the amyloid-PET processing can be found on p.17.

2) It may be better to define the cutoff of Global amyloid PET SUVR by using the whole ADNI sample rather than using only those subjects included in this study. For comparison, the cutoff 1.11 of AV45 suggested by ADNI PET core group may be also used.

Response: We followed the reviewer's suggestion and changed the cut-off for amyloid-positivity to SUVR[FBB] \geq 1.11 and adopted a previously established cut-off for FBB (i.e. SUVR[FBB] \geq 1.08 (see "ADNI_UCBERKELEY_AV45_Methods_12.03.15.pdf" and "UCBerkeley_FBB_Methods_04.11.19.pdf" on the ADNI website). We repeated our mediation analysis in the newly defined group of A β positive subjects. The results remained significant showing that KL-VS^{het} was associated with better ADNI-MEM scores (beta = 0.13, p = 0.040, Cohen's f = 0.104, N = 229; **Fig. 3**) and this association was mediated by lower global tau-PET levels (bootstrapped average causal mediation effect: beta = 0.06, 95% confidence interval = 0.01 - 0.12, p = 0.013, N = 229; **Fig. 4**). We updated the results (p.8) and figures (**Fig. 3, 4**) accordingly.

3) Did you check the normality of the data in this study? If so, please explain the approach for normality test in "Statistical analysis".

Response: We thank the reviewer for raising this important point. The tau-PET SUVR values are usually left-skewed and therefore we entered baseline tau-PET SUVR values as log-transformed

scores into the statistical models in order to approximate normality, a strategy commonly adopted in PET studies^{1,2}. We additionally ensured that our results are not driven by the distribution of the data or outliers. We iteratively determined the t-statistic of the KL-VS^{het} x amyloid interaction effect on tau-PET levels using 1,000 bootstrapping iterations (i.e. random sampling from the subject pool with replacement). The resulting distribution of t-values was significantly greater than zero (inferior temporal ROI: $t(999)=-77.76$, $p<0.001$; global ROI: $t(999)=-83.88$, $p<0.001$) and the 95% confidence intervals did not include zero (inferior temporal ROI: 95% CI = [-4.847,-0.563]; global ROI: 95% CI = [-4.763,-0.794]). This suggests a robust KL-VS^{het} x amyloid interaction effect on tau-PET levels. Next, we repeated the bootstrapping procedure, but randomly reshuffled the KL-VS^{het} labels at every iteration to create a null distribution as a reference. The resulting distribution of t-values was not different from zero when tested by a one-sample t-test (inferior temporal ROI: $t(999)=-1.25$, $p=0.212$; global ROI: $t(999)=-1.065$, $p=0.287$) and given the 95% confidence intervals (CI) (inferior temporal ROI: 95% CI = [-2.670,2.564]; global ROI: 95% CI = [-2.680,2.590]). Importantly, the mean t-value of our bootstrapped distribution of the interaction effect differed significantly from that of the null-distribution (inferior temporal ROI: $t(1901.9)=-47.43$, $p<0.001$; global ROI: $t(1839.2)=-48.99$, $p<0.001$). We included this analysis in the Results on p.6 and in **Supplementary Fig. 2**:

4) In line 397, “To account for potential influences of clinical diagnoses, we ran the same analyses in the subsample of only MCI patients.” What about CN group? It may be informative to include the results of CN group in supplemental material.

Response: Following the reviewer’s suggestions we conducted the regression analysis on the KL-VS genotype x amyloid PET interaction in the CN group separately. We did not observe an interaction effect of KL-VS^{het} x amyloid PET on tau in CN participants (inferior temporal ROI: $\beta = -0.05$, $p = 0.532$, $N = 347$; global ROI: $\beta = -0.05$, $p = 0.539$, $N = 347$; **Supplementary Fig. 1e, f**). However, the interaction effect of KL-VS x amyloid PET on tau was significant in MCI (see above). The absence of a significant effect in the CN group could potentially be due to a stage dependent beneficial effect of Klotho, or alternatively to the fact that the levels of both amyloid and tau PET are lower in CN compared to those in MCI rendering any protective effect harder to detect. We described these results on p.6 and discussed these two alternative interpretations in the Discussion (p. 13).

5) The authors only analyzed cross-sectional tau PET. What about doing the analyses for longitudinal tau changes in amyloid positive individuals or the whole sample (the sample size may be limited for amyloid positive individuals only). As far as I know, at least 185 ADNI participants had baseline amyloid PET, tau PET and longitudinal tau PET within one year according to one latest ADNI study (Guo et al. Biological Psychiatry, 2020, 10.1016/j.biopsych.2020.06.029). This study supports that amyloid pathology drives tau accumulation on the AD continuum, which may be

useful for “Introduction” and “Discussion” of this paper as well.

Response: We agree with the reviewer that it would be very interesting to investigate KL-VS^{het} effects on tau-PET accumulation over time. We selected 200 participants based on the required availability of KL-VS genotype, cross-sectional tau- and amyloid-PET assessed at the same study visit plus an additional follow-up tau-PET scan. The time interval between tau-PET scans was 1.63 years on average (range: 1-4 years). We calculated annual tau-PET SUVR change rates as the difference in tau PET SUVR between the two scans divided by time between scans. Linear multiple regression analysis was used to estimate the KL-VS^{het} x global amyloid SUVR interaction effect on tau-PET annual change rates controlling for age, sex, and diagnosis. We found a significant interaction effect on tau-PET annual change rates in the inferior temporal ROI (beta = -0.22, p = 0.039, N = 200, Cohen’s f = 0.148), but not in the global ROI (beta = -0.15, p = 0.176, N = 200, Cohen’s f = 0.098). KL-VS^{het} carriers showed lower rate of increase in tau-PET in inferior temporal cortices per unit of global amyloid-PET. Thus, the findings support our cross-sectional result, suggesting that the KL-VS^{het} variant is associated with lower amyloid-related tau-PET accumulation. Observing no interaction effect for the global ROI may stem from relatively small annual change rates in tau-PET when averaged across the whole brain over an observation interval of 1.63 years. Indeed, tau disposition in temporal brain regions, including inferior temporal gyri, has been commonly used in longitudinal studies investigating AD-related tau accumulation^{1,2}. We describe these findings on p.6 and in **Fig. 1c, d**. Additionally, we updated the references to include Guo et al., Biological Psychiatry, 2020 since we agree with the reviewer that this paper presents important longitudinal evidence for amyloid-dependent tau accumulation. We added the paper in the Introduction on p.3, l.71 and in the Discussion on p.12 l.282-84.

6)Regarding to the cognition, it may be informative to include other domains of cognitive score provide in ADNI, such as, executive function, language and visuospatial function. As far as I know, they are all available as well as memory. So, it would be very easy to do the analyses of KL-VS, tau and cognition in line 168.

Response: The reviewer raised an interesting point. Since memory is the cognitive domain, which is the earliest affected in AD, we focused on investigating beneficial effects of KL-VS^{het} on ADNI-MEM scores via lower tau-PET accumulation. In secondary explorative analysis, we followed the reviewer’s suggestion and examined the effect on other cognitive domains including executive functions (composite score ADNI-EF), language (composite score ADNI-LAN) and visual spatial perception (composite score ADNI-VS). In amyloid-positive participants, we found KL-VS^{het} to be associated with higher ADNI-LAN scores (beta = 0.14, p = 0.027, Cohen’s f = 0.114, N = 229; **Supplementary Fig. 4a**), a trend-level association with higher ADNI-EF scores (beta = 0.12, p = 0.067, Cohen’s f = 0.095, N = 229) and no association with ADNI-VS scores (beta = 0.05, p = 0.468, N = 229). The beneficial effect of KL-VS^{het} on language abilities was mediated by lower global tau-PET levels in KL-VS^{het} carriers versus non-carriers (bootstrapped average causal mediation effect: beta = 0.05, 95% confidence interval = 0.01 - 0.12, p = 0.017, N = 229; **Supplementary Fig 4b**). We added these results on p.9.

7)Since the authors only observed predictive effect of tau aggregation in amyloid positive individuals, can you comment why no significant effect was found in the whole sample or amyloid negative individuals?

Response: This is an interesting question. The moderation of the association between amyloid and tau pathology is probably the statistically more powerful approach. Consistent with the results from Guo et al. mentioned above, our study design was based on the assumption that A β is the key driving force of tau-PET and any beneficial effects should be best detected as an attenuation of the association between A β and tau pathology. Currently, little is known about the molecular mechanism that renders A β the driving force of tau pathology and how Klotho may modify such a process. Our results encourage future studies to investigate potential mechanisms.

Reviewer #2 (Remarks to the Author):

Summary:

The authors investigated the role of Klotho-VS heterozygosity (KL-VShet) carriership within the context of Alzheimer's disease. While previous work already showed that carriership was associated with reduced levels of amyloid deposition in APOE- ϵ 4 carriers, the current paper focused on downstream effects, namely tau accumulation and cognitive performance. The authors show that in elderly subjects with elevated amyloid burden, KL-VShet carriership was associated to lower tau burden as measured with PET imaging. In addition, there seemed to be a local association, as the spatial analyses revealed that Klotho was most strongly associated with temporal tau burden. Finally, KL-VShet carriership was associated with better memory performance, which was mediated by tau-PET levels, further supporting the influence of this gene on tau pathology.

The paper is novel, includes a respectable sample size, and is complete in terms of statistical analyses. In general, the paper is well written, but the methods section requires some clarification.

Major comments:

1) The methods section, specifically the statistical analyses subsection, is sometimes unclear and confusing. I suggest thorough rewriting and maybe the use of subheadings. Specific comments are:
- **Sample, page 14, line 325-328: Please add the N of each included Dx subgroup.**

Response: As suggested, we inserted subheadings to improve comprehensibility. Whenever applicable, we divided the analysis into a "main analysis" and "secondary analyses". We refer to these subdivisions also when reporting the results in the "Results" section of the manuscript. Important information that applies to all statistical models is now pointed out at the beginning of the "Statistical Analysis" paragraph (see p.19).

2) MR and PET acquisition and preprocessing, page 15, line 341: The authors state '50-70 min post-injection of 18F-AV45 Florbetaben'. There are several possible mistakes here. Was the tracer used Florbetaben? Then it should not state AV45 (which is Florbetapir) and the acquisition interval should be 90-110 p.i.. If indeed Florbetaben was used, then I hope that the correct interval (i.e. 90-110 p.i.) was used to obtain the SUVR data, as in accordance with the ADNI processing and product manufacturer guidelines. Please check and correct.

Response: we thank the reviewer for pointing that out. We clarify that Florbetapir (FBP) rather than Florbetaben (FBB) amyloid-PET was used in the original version of the manuscript. We corrected that mistake. Please note that in response to the suggestion of reviewer 1, we added FBB amyloid-PET scans that became recently available in ADNI3, and which were not included in our submitted data analysis. Given that our inclusion criteria were the availability of amyloid-PET and tau-PET scans (among others) the inclusion of subjects with FBB scans plus tau-PET increased our sample from 357 to 551 participants. In order to make data from the two PET-tracers comparable, we transformed SUVR values into centiloid units using the established transformation formula by Jagust and Landau http://adni.loni.usc.edu/wp-content/themes/freshnews-dev-v2/documents/pet/ADNI_Centiloids_Final.pdf. We updated the description of the amyloid-PET preprocessing on p.16. Then, we reran all statistical analyses in the updated larger sample. Our results remained comparable to the ones reported for the original sample (p. 5-10 in manuscript, see also response to comment 1 by reviewer 1).

3) Statistical analyses, page 17/18: The analyses described are jumping back and forth regarding the selection and use of subsamples. For example, in line 397 it is stated that the analyses was rerun only in MCI patients. Subsequently the next sentence starts with 'Additional'; was this analyses then done in the whole sample, or in the previously described MCI subsample? Please

improve clarity. Also, in line 399-403 first a stratified analyses is described followed by the analyses in the whole cohort. This is also a bit of a confusing order to list step-wise analyses.

Response: We thank the reviewer for this helpful comment. We changed the order of the analyses to start with examinations in the whole sample, before stratified analyses are reported. The only exception to this rule was made for the question of whether KL-VS^{het} is associated with better memory via lowering tau-PET levels. Here, mediation analysis in the subsample of amyloid-positive participants is in our opinion the preferable approach compared to the secondary analysis including the interaction analysis in the whole sample. Hence, we firstly reported the mediation analysis in a subsample before the whole cohort interaction analysis. To additionally improve the understanding of which subjects were included in a particular analysis, we report the Ns for each analysis in both the “Statistical Analysis” subsection and the “Results” section. Please see p.19 and the following for the updated report of the statistical analysis.

4) Statistical analyses, page 18, lines 418-420: It is unclear whether only subjects with A β + status were thus included for this sub-analyses. Also, please specify how the optimal threshold was determined in addition to referencing the supplementary material as this is quite key in an early population.

Response: The reviewer raised an important point that was also noted by other reviewers. Following their suggestions, elevated amyloid-PET uptake was determined according to the established cut-off values, i.e. SUVR[FBP] \geq 1.11 or SUVR[FBB] \geq 1.08 (“ADNI_UCBERKELEY_AV45_Methods_12.03.15.pdf” and “UCBerkeley_FBB_Methods_04.11.19.pdf” on the ADNI website), and not using a study-specific threshold. Hence, only participants with an A β + status were included into the mediation analysis (N = 229). Consistent with the original results, we found a significant mediation effect suggesting that KL-VS^{het} is associated with better ADNI-MEM scores (beta = 0.13, p = 0.040, Cohen’s f = 0.104, N = 229; **Fig. 3**) and this effect is mediated by lower global tau-PET levels in this group (bootstrapped average causal mediation effect: beta = 0.06, 95% confidence interval = 0.01 - 0.12, p = 0.013; **Fig. 4**). We modified our results (p.8) and figures (**Fig. 3, 4**) accordingly.

5) Statistical analyses, page 18, line 423: was the interaction analyses presented here with continuous amyloid-PET or with A β status?

Response: This analysis was originally been performed with continuous amyloid-PET SUVR values. After rerunning the analysis in the updated larger sample using continuous FBP and FBB amyloid-PET CL units, we found a significant KL-VS^{het} x amyloid-PET interaction effect on memory functions (beta = 0.08, p = 0.037, Cohen’s f = 0.090, N = 549). Specifically, we observed that individuals with high amyloid-PET burden showed better memory performance when being KL-VS^{het} carriers compared to non-carriers (**Supplementary Fig. 4**). This result is reported on p.9.

6) Throughout: add the N when any subgroup is mentioned.

Response: We agree and added Ns for each analysis.

Additional comments:

7) Results, page 5, line 120: the finding that the KL-VShet was protective against A β -associated accumulation of tau pathology was also repeated in the MCI cohort. Considering that the tau-PET ROI used in this work was aimed to be sensitive of early pathology, it would be interesting the repeat the analyses also in the CN population. The sample size of this diagnostic group would allow for such stratification.

Response: We thank the reviewer for this suggestion which had been also made by other reviewers. We updated the manuscript including the interaction analysis in the CN subgroup on p.6 which yielded no KL-VS^{het} x amyloid-PET interaction effects on tau-PET levels in either ROI (both p>0.05; Supplementary Fig 1e, f). Please see our response to comment 4 of reviewer 1 for more details.

8) Discussion, page 12, line 278-286: the spatial correspondence is discussed in this paragraph of the discussion section. However, I feel that one important aspect is missing and should be considered as an explanation; considering the relatively early population used in this work (which is an asset!), the finding of an association between Klotho and tau pathology in temporal brain areas, could be due to a power issue, as more wide-spread cortical tau is probably less frequently observed in this population. Please add this as a consideration when interpreting these results.

Response: The reviewer makes an important point. From previous investigations it is known that temporal brain areas and especially the inferior temporal cortex (ITC) are among the first to show amyloid-related increases of tau-PET levels¹⁻⁵. Therefore, we a priori selected ITC as one ROI for which we expected to observe a strong relationship between tau- and amyloid-PET levels and which is hence very well suited for investigating a potentially moderating role of KL-VShet status on amyloid-tau relationship. In addition, we chose a global ROI in order to capture more wide-spread effects of KL-VShet on amyloid-related tau pathology. Since we found a KL-VShet x amyloid-PET effect on tau-PET levels in both the inferior temporal (beta = -0.12, p = 0.009, N = 551, Cohen's f = 0.112; **Fig. 1a**) and global ROI (beta = -0.13, p = 0.008, N=551, Cohen's f = 0.114; **Fig. 1b**) significant, KL-VShet effects seem to be not restricted to early tau regions.

9) Discussion, page 13, line 306-307: the sentence 'In summary, due to the high clinical relevance of tau accumulation and mixed results from trials on Aβ immunization, strategies for targeting tau pathology are moving into focus.' Is a weird sentence to start the conclusion with, especially considering that after the words 'In summary' you don't summarize your own results. Also, the utility of this work in the context of trials is not discussed at all, which I would opt for including in the paper.

Response: We agree to the comment and changed this sentence to "In summary, our findings revealed a protective effect of KL-VShet against tau accumulation which particularly manifested in amyloid-positive individuals, where lower tau pathology was related to better cognitive functions." (p.15).

Reviewer #3 (Remarks to the Author):

In this study, Neitzel and colleagues sought to determine whether KL-VShet is associated with lower levels of pathologic tau and whether KL-VShet was also associated with lower memory impairment. For this purpose, they studied 354 individuals including CN, MCI, and AD; of whom 92 were KL-VShet and 262 were KL-VS noncarriers. They found that KL-VShet showed lower cross-sectional increase in tau-PET per unit increase in amyloid-PET when compared to non-carriers. This effect of KL-VShet on tau-PET also seemed stronger in Klotho mRNA-expressing brain regions mapped onto a gene expression atlas. Lastly, KL-VShet was related to better memory functions and this association was mediated by lower tau-PET.

This is overall a well-written paper. Some comments for the authors' consideration follow:

1. As specified in the title and other sections, the paper's purported objective is to test whether KL-VShet is associated with reduced tau. This however is not what was actually done. They authors instead examined whether KL-VShet modifies the association between abeta and tau. Recommend either re-aligning analyses with purported aim or rewording the title (and all other affected sections) to align with what was actually done

Response: We thank the review for this important comment. The main objective of the study was to investigate whether KL-VS^{het} modifies the relationship between amyloid- and tau-PET, since amyloid is the key driving force of tau accumulation in AD. This analysis strategy allowed us to take stage-dependent tau-PET levels into account, i.e. is the increase in tau-PET as a function of the amyloid level lower in KL-VS^{het} carriers vs non-carriers, which increased the statistical power to detect Klotho related effects on tau-PET. Following the reviewer's suggestion, we reworded the title to better reflect the study's key finding into "KL-VS heterozygosity modifies amyloid-dependent tau accumulation and memory impairment in Alzheimer's disease". Moreover, we rephrased the manuscript to more clearly state that the KL-VS^{het} effect on tau accumulation was dependent on amyloid-PET levels as follows (updates are underscored):

Abstract (p.2): "Together, our findings provide evidence for a protective role of KL-VS^{het} against amyloid-related tau pathology and tau-related memory impairments in elderly humans at risk of AD dementia"

Introduction (p.4): "Overall, we show a protective role of KL-VShet against the development of AD-related tau pathology and thus cognitive decline, suggesting that Klotho could be an attractive treatment target to slow the progression of AD."

Discussion (p.12): In amyloid-positive participants, the KL-VS^{het} variant was associated with better memory performance, and this relationship was mediated by lower tau-PET levels, suggesting that lower levels of pathologic tau in the KL-VS^{het} carriers explain the protective association between KL-VS^{het} and memory performance. Although our findings do not implicate a causative mechanism of Klotho in AD, we provide evidence for a protective role of KL-VS^{het} against amyloid-dependent tau pathology which is the key AD brain alteration linked to cognitive impairment.

Discussion (p.14): "Those mechanistic explanations remain speculative at this point and the current work encourages future studies to investigate the mechanism that could underlie the protection Klotho exerts against the development of AD-related tau pathology."

2. The analyses, as stated above, are premised on KL-VShet modifying the association between abeta and tau. However, the authors reported (Table 1) that the KL-VS groups do not differ on abeta levels. I believe this has considerable implications for the rest of the models

Response: It is correct that observing no significant difference in amyloid-PET levels in KL-VS^{het} carriers versus non-carriers in the tau-PET sample (N = 551) has an important implication for the interpretation of our findings. It specifically suggests that the beneficial effect of KL-VS^{het} on tau-PET levels is not mediated by the effect of KL-VS^{het} on amyloid-PET burden. Rather, KL-VS^{het} modifies the association between amyloid and tau.

3. The PET data appear to include several cases where SUVR is less than 1. Such data are functionally meaningless, and I suggest removing them from the analyses

Response: The reviewer is correct that some participants showed a tau-PET SUVR slightly smaller than 1, i.e. 2 participants for the inferior temporal and 51 out of 551 participants for the global ROI. This can happen when tau-PET levels are not elevated above the tau-PET level of the reference region (inferior cerebellar grey matter). When excluding these participants, we still found a significant KL-VS^{het} x amyloid interaction effect of tau-PET levels (inferior temporal ROI: beta = -0.12, p = 0.007, N = 549; global ROI: beta = -0.15, p = 0.003, N = 500). Given that such low values may simply indicate low levels of tau-PET, but do not necessarily reflect measurement error (there is no absolute 0 value for the PET signal), we refrained from excluding those values from the overall analysis. However, we agree that the evaluation of a potential influence of such values is important and included the sub-analysis on p.6 and **Supplementary Fig. 1g, f**.

4. Similarly, to describe “elevated amyloid burden” as an SUVR of >1 is neither acceptable nor in line with the extensive body of work in this area. If the authors would like to retain this analysis in the paper, they are encouraged to recalibrate their definition of elevated abeta.

Response: We thank the reviewer for this important remark. Please note that in response to the suggestion of reviewer 1, we added Florbetaben (FBB) besides Florbetaphir (FBP) amyloid-PET scans that became recently available in ADNI3, but which were not included in our original data analysis. This increased our sample from 357 to 551 participants (please see response to comment 1 of reviewer 1). We agree to the comment about the amyloid-PET SUVR threshold and changed the cut-off for sample selection to the established cut-off for amyloid-positivity, i.e. SUVR[FBP] ≥ 1.11 or SUVR[FBB] ≥ 1.08. Then, we reran the mediation analysis in only amyloid-positive participants (N = 229) which yielded comparable results to the ones originally reported (please see response to comment 2 of reviewer 1). We modified our results (p.8) and figures (**Fig. 3, 4**) accordingly.

5. Given the structure of the paper (that the findings from the tau model tie back into the memory model), it’s concerning that the tau model and the memory model are not fit in concordance with each other. As mentioned above, the tau model is about KL-VShet modifying the association between abeta and tau whereas the memory model is about KL-VShet directly leading to better memory scores

Response: We investigated the question of whether the KL-VS^{het} variant is associated with better memory via lowering tau- PET levels *dependent* on Aβ levels. Note that we conducted a mediation analysis of the effect of KL-VS^{het} on ADNI-MEM with tau-PET levels as mediator, where the analysis was stratified by amyloid PET status. Based on the rationale that the effect of KL-VS on tau PET is dependent on abnormal Aβ levels, we tested the mediation analysis in the Aβ positive subjects, where the mediation model was statistically significant. A mediation effect in the Aβ negative sample was not evident, because no KL-VS effect on tau PET was present.

6. It is customary when reporting a significant interaction (eg of KL-VS and abeta on tau) to follow it up by reporting on the simple slopes (eg the association of abeta and tau in the two KL-VS groups). This was not done in this case

Response: We reported the results of the slopes of tau-PET as a function of global amyloid-PET for KL-VS^{het} carriers and non-carriers for the main analyses in **Supplementary Table 1**.

7. The sample sizes in the KL-VShet vs KL-VS noncarrier groups are markedly disproportionate. This raises the possibility that the lack of association in the KL-VShet group is driven by the smaller sample. It would be important to reassure readers that this was not the case. One approach to doing this would be to use a matching algorithm to identify 92 KL-VS noncarriers who are matched to the 92 KL-VShet on relevant covariates; then repeat the analyses in this matched sample

Response: We thank the reviewer for this suggestion. We followed the recommendation and selected 144 out of 407 KL-VS^{het} non-carriers based on propensity score matching for global amyloid-PET levels and diagnosis (not that there are now more KL-VS^{het} carriers due to the increased sample size included in the revised manuscript, see also response to comment 1 by reviewer 1). Comparable KL-VS^{het} x amyloid-PET interaction effects were found on tau-PET levels in both ROIs (inferior temporal: beta = -0.20, p = 0.005, N = 288, Cohen's f = 0.168 global: beta = -0.21, p = 0.010, N = 288, Cohen's f = 0.156; **Supplementary Fig. 1a, b**). This result suggests that in KL-VS^{het} carriers the A β -associated accumulation of tau pathology was attenuated, which could not be attributed to a potential bias by unequal sample sizes of the genotype groups. All analyses were controlled for main effects of the interaction terms, age, sex, diagnosis, education and ApoE ϵ 4 carrier status. We report this finding on p.5.

8. The inclusion of all three diagnostic groups in the analyses exposes the models and conclusion to muddling. Suggest either focusing on only one diagnostic group or running the analyses separately in each of the diagnostic groups

Response: Considering our main aim to investigate the interaction between KL-VS^{het} and amyloid burden on tau-PET levels, the inclusion of a wider clinical range provides more statistical power and is feasible from the point of view that AD pathology develops in a continuous fashion. Pooling across different diagnosis groups has been also adopted by other studies investigating genetic effects on tau pathology^{6,7,8}. Hence, we decided to perform our analyses first in the whole sample controlling for diagnosis as a covariate and subsequently report stratified analyses in different diagnosis subgroups. In response to the reviewer's comment and suggestions by other reviewers, we additionally included the investigation in the CN subgroup on p.6 and **Supplementary Fig. 1e, f**. While we found significantly lower amyloid-dependent tau-PET levels in MCI KL-VS^{het} carriers versus non-carriers, we did not observe a beneficial effect of KL-VS^{het} in CN participants. Please see response to comment 4 of reviewer 1. Due to the low sample size of dementia patients (N = 48), we did not perform a stratified analysis in this clinical group. We prefer to keep the pooled analysis along with the subgroup analysis in order to render the full information for the reader.

9. The purpose and significance of the mRNA analyses was unclear to me. Importantly, p of .05 is inappropriate for whole brain analyses, even under exploratory conditions

Response: In this analysis we compared the strength of the KL-VS^{het} x amyloid interaction effect on tau-PET levels (t-values) with the strength of the mRNA expression (log2 values) using one correlation analysis (r=0.46, p=0.007, CI[95%]=0.23-0.68). Since only one test was performed, it is not necessary to correct for multiple testing. For illustration purpose only, we showed the KL-VS^{het} x amyloid interaction effect on tau-PET levels mapped onto the Freesurfer atlas in Figure 2a. In Figure 2b we originally thresholded the map at p<0.05 to facilitate visual comparability with mRNA expression. In response to the reviewer's comment, we now thresholded the map at p<0.01. Yet, t-values from all Freesurfer regions (not only the significant ones) were included in the correlation analysis with mRNA expression. For better understanding, we now explicitly state that interaction test statistics for all regions were considered on p.20: "We then tested the ROI-to-ROI Pearson-Moment correlation between regional KL mRNA expression and the interaction effect test statistic (not restricted to regions showing a significant interaction effect)." We also re-worded the description of the Results on p. 7, line 172-73 as follows "In order to estimate the spatial overlap between the strength of KL gene expression and the test statistic of the KL-VS^{het} x amyloid-PET interaction on tau-PET, [...]"

References

1. Guo, T., Korman, D., Baker, S. L., Landau, S. M. & Jagust, W. J. Longitudinal Cognitive and Biomarker Measurements Support a Unidirectional Pathway in Alzheimer's Disease Pathophysiology. *Biol. Psychiatry* (2020) doi:<https://doi.org/10.1016/j.biopsych.2020.06.029>.

2. Jack Jr, C. R. *et al.* Predicting future rates of tau accumulation on PET. *Brain* **143**, 3136–3150 (2020).
3. Pontecorvo, M. J. *et al.* Relationships between flortaucipir PET tau binding and amyloid burden, clinical diagnosis, age and cognition. *Brain* **140**, 748–763 (2017).
4. Lockhart, S. N. *et al.* Amyloid and tau PET demonstrate region-specific associations in normal older people. *Neuroimage* **150**, 191–199 (2017).
5. Sepulcre, J. *et al.* In vivo tau, amyloid, and gray matter profiles in the aging brain. *J. Neurosci.* **36**, 7364–7374 (2016).
6. Hohman, T. J. *et al.* Sex-specific association of apolipoprotein E with cerebrospinal fluid levels of tau. *JAMA Neurol.* **75**, 989–998 (2018).
7. Buckley, R. F. *et al.* Sex mediates relationships between regional tau pathology and cognitive decline. *Ann. Neurol.* (2020).
8. Therriault, J. *et al.* APOE ϵ 4 potentiates the relationship between amyloid- β and tau pathologies. *Mol. Psychiatry* 1–12 (2020).

REVIEWER COMMENTS

Reviewer #1 (Remarks to the Author):

The authors had answered all my concerns and updated the manuscript accordingly. Just want to remind the authors again, florbetapir PET signal in ADNI was acquired between 50-70 min post-injection (line 391), please updated this. I have no more comments.

Reviewer #2 (Remarks to the Author):

The authors reponded adequately to all comments and made the appropriate changes to the manuscript.

The only thing still missing is a paragraph in the discussion section on how these results are of importance for clinical trials, e.g. in which setting, study design, etc., especially considering that this is stated in the concluding final paragraph of this section, but is not discussed in more depth. In my opinion, this would further add to the importance of the work.

Reviewer #4 (Remarks to the Author):

It is reported that Klotho is associated with enhanced longevity and better brain health in aging. Authors explored the correlation between KL-VS heterozygosity (KL-VShet) and brain tau pathology in a cohort of older asymptomatic and symptomatic individuals. They found that KL-VShet variant was related to an attenuated increase in regional tau PET uptake at pathological levels of global amyloid PET. They also found that KL-VShet was associated with better memory performance, which was mediated by reduced tau-PET levels. Thus, they made a point that KL-VShet variant was protective against AD-related increase in neurofibrillary tangles.

My comments are as follows:

1. Authors made the conclusions in the third paragraph of the Introduction section. It could be more convincing if they put the concluding statements in the discussion section.
2. In line 110, authors mentioned that they focus on tau PET in the inferior temporal cortex which was reported to be the ROI of early A β -related tau pathology. However, they also concluded that KL-VShet variant was protective against AD-related increase in neurofibrillary tangles. As ADNI provided tau PET data of subdivided brain regions, such as the entorhinal cortex (Braak 1) and the hippocampal cortex (Braak 2). It was better to evaluate the possible correlation on precise ROIs rather than merely the inferior temporal cortex and the whole.
3. In line 156-159, authors mentioned: "KL-VShet carriers showed lower tau-PET increases in inferior temporal cortices over time as a function of rising global amyloid-PET levels (Fig. 1c, d) suggesting that the KL-VShet variant was protective against A β -associated spread of tau pathology." Were there statistical differences in the longitudinal change rates of tau PET between these two groups? It was not shown in Fig. 1c, d or the text word.
4. Authors need better explain the "A β - associated spread of tau pathology". As they merely assessed tau PET of the inferior temporal cortex and the whole brain, it was unreasonable to use the "spread of tau pathology".

Reviewer #1 (Remarks to the Author):

The authors had answered all my concerns and updated the manuscript accordingly. Just want to remind the authors again, florbetapir PET signal in ADNI was acquired between 50-70 min post-injection (line 391), please updated this. I have no more comments.

Response: thank you, we updated the sentence as suggested "Amyloid-PET scans were obtained during 4x5 min time frames measured 50-70 min post-injection of 18F-FBP or 90-110 min post-injection of 18F-FBB."

Reviewer #2 (Remarks to the Author):

The authors reponded adequately to all comments and made the appropriate changes to the manuscript.

The only thing still missing is a paragraph in the discussion section on how these results are of importance for clinical trials, e.g. in which setting, study design, etc., especially considering that this is stated in the concluding final paragraph of this section, but is not discussed in more depth. In my opinion, this would further add to the importance of the work.

Response: We followed the reviewer's suggestion and added a short paragraph about importance for clinical trials in the discussion (line 338-347):

"Our results have important implications for clinical trials in AD. Since tau pathology correlates more closely with clinical symptoms than A β , tau-targeted therapies seem a promising approach to arrest disease progression¹. The common KL-VS genotype may inform those clinical trials that target tau pathology. Especially when anti-tau trials aim to include A β + or ApoE4+ participants, group differences in the KL-VS^{het} variant may be taken into account when estimating the expected change in tau pathology over time, which would be useful in the computation of statistical power to detect a treatment effect. Furthermore, the current findings encourage future studies to test whether enhancing Klotho protein levels could reduce the development of tau pathology in A β + participants. The Klotho protein is druggable and could thus be made a target in the development of disease-modifying therapeutic approaches."

Reviewer #4 (Remarks to the Author):

It is reported that Klotho is associated with enhanced longevity and better brain health in aging. Authors explored the correlation between KL-VS heterozygosity (KL-VShet) and brain tau pathology in a cohort of older asymptomatic and symptomatic individuals. They found that KL-VShet variant was related to an attenuated increase in regional tau PET uptake at pathological levels of global amyloid PET. They also found that KL-VShet was associated with better memory performance, which was mediated by reduced tau-PET levels. Thus, they made a point that KL-VShet variant was protective against AD-related increase in neurofibrillary tangles.

My comments are as follows:

1. Authors made the conclusions in the third paragraph of the Introduction section. It could be more convincing if they put the concluding statements in the discussion section.

Response: As suggested by the reviewed, we removed the concluding statement from the introduction.

2. In line 110, authors mentioned that they focus on tau PET in the inferior temporal cortex which was reported to be the ROI of early A β -related tau pathology. However, they also concluded that KL-VShet variant was protective against AD-related increase in neurofibrillary tangles. As ADNI provided tau PET data of subdivided brain regions, such as the entorhinal cortex (Braak 1) and the hippocampal cortex (Braak 2). It was better to evaluate the possible correlation on precise ROIs rather than merely the inferior temporal cortex and the whole.

Response: We agree that region-specific KL-VS^{het} x amyloid interaction effects are interesting to explore. However, we deliberately focused in the primary analysis on a hypothesis-driven set of tau PET ROIs in order to reduce the risk of Type I error. As mentioned in our manuscript we focused on the inferior temporal cortex to assess early tau PET accumulation in AD. The inferior temporal region shows increased tau PET accumulation specifically in subjects with biomarker evidence of A β pathology, i.e. a defining pathology of AD^{2, 3-6}, and is thus one of the earliest regions showing AD related tau PET increase. In addition, we focused on global tau PET to reflect brain wide tau PET uptake given that there was no reason a priori to hypothesis regional differences in the effect of KL-VS on tau PET.

However, in a secondary exploratory analysis, we did conduct a ROI-wise analysis (see Figure 2). In response to the reviewer's comment, we included Supplementary Table 2 which presents the detailed statistical results of the ROI-wise interaction analyses. Out of the 34 left-hemispheric, cortical Freesurfer atlas regions, 18 regions showed a significant KL-VS^{het} x amyloid PET interaction effect at $P < 0.01$, including the entorhinal cortex (see Figure 2b). These regions were primarily located in the temporal and frontal lobes as well as in the precuneus, while primary motor and sensory cortices were spared. With regard to Braak tau-staging, we found a significant interaction effect in Braak I (entorhinal cortex) and most Braak III regions (parahippocampal and fusiform gyri, but not lingual gyrus) as well as in many Braak IV (e.g. inferior and middle temporal gyri) and Braak V regions (e.g. superior temporal and frontal gyri). No effect was found in Braak VI regions. Note that we refrained from investigating hippocampal Flortaucipir-PET (Braak II), since tracer off-target binding has been reported in the hippocampus⁷.

3. In line 156-159, authors mentioned: "KL-VShet carriers showed lower tau-PET increases in inferior temporal cortices over time as a function of rising global amyloid-PET levels (Fig. 1c, d) suggesting that the KL-VShet variant was protective against A β -associated spread of tau pathology." Were there statistical differences in the longitudinal change rates of tau PET between these two groups? It was not shown in Fig. 1c, d or the text word.

Response: In response to the reviewer's comment, we investigated whether beyond an interaction effect of KL-VS x amyloid PET also a main effect of KL-VS^{het} variant on tau-PET annual change rates (independent of A β pathology) was present. We did not observe a main effect (β =-0.033, SE=0.006, P=0.63). This finding suggests that KL-VS^{het} may modulate processes downstream of A β resulting in enhanced tau accumulation.

4. Authors need better explain the “A β - associated spread of tau pathology”. As they merely assessed tau PET of the inferior temporal cortex and the whole brain, it was unreasonable to use the “spread of tau pathology”.

Response: We agree with the reviewer and changed the wording to A β -related ‘increase’ of tau pathology rather than “spreading” throughout the manuscript.

References

1. Congdon, E. E. & Sigurdsson, E. M. Tau-targeting therapies for Alzheimer disease. *Nat. Rev. Neurol.* **14**, 399–415 (2018).
2. Pontecorvo, M. J. *et al.* Relationships between flortaucipir PET tau binding and amyloid burden, clinical diagnosis, age and cognition. *Brain* **140**, 748–763 (2017).
3. Johnson, K. A. *et al.* Tau positron emission tomographic imaging in aging and early Alzheimer disease. *Ann. Neurol.* **79**, 110–119 (2016).
4. Sepulcre, J. *et al.* In vivo tau, amyloid, and gray matter profiles in the aging brain. *J. Neurosci.* **36**, 7364–7374 (2016).
5. Schöll, M. *et al.* PET imaging of tau deposition in the aging human brain. *Neuron* **89**, 971–982 (2016).
6. Lockhart, S. N. *et al.* Amyloid and tau PET demonstrate region-specific associations in normal older people. *Neuroimage* **150**, 191–199 (2017).
7. Marquie, M. *et al.* Validating novel tau positron emission tomography tracer [F-18]-AV-1451 (T807) on postmortem brain tissue. *Ann. Neurol.* **78**, 787–800 (2015).

REVIEWER COMMENTS

Reviewer #2 (Remarks to the Author):

The authors have added an adequate discussion paragraph on the value of their results for clinical trials and therefore have fully addressed my remaining comment.

Reviewer #4 (Remarks to the Author):

The authors responded to the concerns and have made corresponding changes to the manuscript. I have no more comments.

Reviewer #2 (Remarks to the Author):

The authors have added an adequate discussion paragraph on the value of their results for clinical trials and therefore have fully addressed my remaining comment.

Response: We thank the reviewer for the thoughtful reviews.

Reviewer #4 (Remarks to the Author):

The authors responded to the concerns and have made corresponding changes to the manuscript. I have no more comments.

Response: We thank the reviewer for the thoughtful review.